# MXene based saturation organic vertical photoelectric transistors with low subthreshold swing

Enlong Li[1,2], Changsong Gao[1,2], Rengjian Yu[1,2], Xiumei Wang[1,2], Lihua He[1,2], Yuanyuan Hu[3], Huajie Chen[4], Huipeng Chen [1,2 ✉] & Tailiang Guo[1,2]

Vertical transistors have attracted enormous attention in the next-generation electronic devices due to their high working frequency, low operation voltage and large current density, while a major scientific and technological challenge for high performance vertical transistor is to find suitable source electrode. Herein, an MXene material, $Ti_3C_2T_x$, is introduced as source electrode of organic vertical transistors. The porous MXene films take the advantage of both partially shielding effect of graphene and the direct modulation of the Schottky barrier at the mesh electrode, which significantly enhances the ability of gate modulation and reduces the subthreshold swing to 73 mV/dec. More importantly, the saturation of output current which is essential for all transistor-based applications but remains a great challenge for vertical transistors, is easily achieved in our device due to the ultra-thin thickness and native oxidation of MXene, as verified by finite-element simulations. Finally, our device also possesses great potential for being used as wide-spectrum photodetector with fast response speed without complex material and structure design. This work demonstrates that MXene as source electrode offers plenty of opportunities for high performance vertical transistors and photoelectric devices.

[1] Institute of Optoelectronic Display, National and Local United Engineering Lab of Flat Panel Display Technology, Fuzhou University, Fuzhou 350002, China. [2] Fujian Science and Technology Innovation Laboratory for Optoelectronic Information of China, Fuzhou 350100, China. [3] College of Semiconductors (College of Integrated Circuits), Hunan University, Changsha 410082, China. [4] College of Chemistry, Xiangtan University, Xiangtan 411105, China. ✉email: hpchen@fzu.edu.cn

Organic field-effect transistors (OFETs) have attracted enormous attentions due to their flexibility, materials diversity, solution-processability and large-scale compatibility[1–4]. However, the features of large current density and fast switching speed are not available in OFETs, which are two key attributes of high-performance transistors. Due to the low mobility of organic semiconductor, an ultra-short channel is usually required for OFET to excel in the mentioned features, while the leakage current increases with decrease of channel length, leading to degradation of device performance[5–7]. An alternative approach is the utilization of vertical organic field-effect transistor (VOFET), in which the semiconductor is sandwiched between source and drain electrode and the current flows vertically through semiconductor layer. Such geometry provides a nanoscale channel length depending on the thickness of the active layer without complicated lithography process, and results in high current density and working frequency under low operating voltage. The continuous improvement of device performance and the wide application of VOFETs in high-frequency electronics[8], photodetectors[9–11], synaptic transistors[12,13] and organic light-emitting transistors[14,15] have proved their great potential in organic electronics.

Rather than direct modulation of carrier density in semiconductor layer as seen in planar OFETs, the operation of VOFET relies on the modulation of Schottky barrier (SB) between source electrode and semiconductor, which dominates the injection of carriers. Therefore, the major scientific and technology challenge for VOFET is to find suitable source electrode which forms excellent contact and appropriate SB with semiconductor, and meanwhile yields weak electric field shielding of electrostatic gating[5,16,17]. In graphene based VOFETs, the graphene only partially screens the gate electric field, thus the gate electric field can penetrate graphene and then modulate the SB. However, the shield effect of graphene still reduces the gate controllability of VOFET[18,19]. In addition to graphene, there are some VOFETs using mesh metal electrode such as Ag nanowires (AgNWs) as the source electrode since the gate field can penetrate the perforation of mesh electrode to directly modulate the SB[5,20,21]. Although the porous structure reduces the shielding effect of gate electric field and improves the gate control over the device, the large sheet resistance and undesirable contact with semiconductor layer also limit the performance of VOFETs based on mesh source electrodes. Besides, the fixed work function of mesh metal and graphene limits their compatibility with many organic materials. Moreover, the saturation of output current which is essential for all transistor-based applications remains a great challenge for VOFETs due to the leakage current between source and drain electrode induced by their vertical stacking in space.

Transition-metal carbides/nitrides (known as MXenes), an emerging family of two-dimensional (2D) materials, have attracted enormous attention since their first discovery in 2011[22]. Their excellent and tailorable electronic properties render them attractive for applications in various electronic devices, such as flexible/transparent electrodes, transistors and sensors[23–25]. The spontaneous oxidation-induced MXene-nano-$TiO_2$ composites also raise significant interest for optoelectronic, energy storage and biomedical applications[26]. Besides, the hydrophilic nature and thermal stability have endowed MXene great potential for fabricating low-cost and large-area solution-processed devices. Meanwhile, their work function can be regulated by terminal ligand from 2.14 eV to over 5.65 eV, providing tunable barrier height with semiconductor materials, which is a potentially important characteristic in Schottky barrier based devices[27–29]. Finally, the single layer MXene film possesses an ultra-thin thickness of about 1 nm which is comparable to that of a single layer graphene and the stacked MXene flakes can form conductive MXene film with perforation. Therefore, MXene is highly attractive for being used as the source electrode of VOFET.

Herein, an MXene material, $Ti_3C_2T_x$, is introduced as the source electrode of VOFET, which is denoted as MVOFET. Benefiting from the ultra-thin thickness and the perforation of MXene film, the MVOFET takes advantage of both SB modulation modes of graphene and mesh metal electrode, which significantly improves the gate controllability of device. The MVOFET exhibits a small subthreshold swing (SS) of 73 mV/dec and a small threshold voltage of −1.2 V, which are remarkably low compared with those of a traditional AgNWs-based vertical transistor. Besides, the unsaturation of output characteristic, which is frequently seen in VOFETs, vanishes due to the ultra-thin thickness and native oxidation of MXene, as proved by COMSOL simulations. Moreover, the MVOFET exhibits a wide-spectrum detection behavior from ultraviolet (UV) to visible light, which enables high photodetection performance and fast response speed under UV (10 ms) and visible light (0.21 s) illumination. Our work shows the feasibility of employing MXenes for high-performance VOFETs and photoelectric devices.

## Results

The chemical structure of organic semiconductors used in this study are shown in Fig. 1a. The MXene solution with different concentration was deposited on Si wafer with 100 nm $SiO_2$ and further characterized by scanning electron microscopy (SEM) (Fig. 1b and Supplementary Fig. 1). As shown in Fig. 1b, the $Ti_3C_2T_x$ film have many perforations which are attributed to random deposition of $Ti_3C_2T_x$ nanosheets and the areas of the perforations can be regulated by the concentration of MXene solution. The continuous conductive channel ensures efficient injection of carriers and the perforations enable gate modulation of device current, which lay the foundation for operation of MVOFET device. The morphology of 2D nanosheet structure was confirmed by transmission electron microscope (TEM). Figure 1c shows the stacking of two nanosheets with different shape and size. Small pin holes can also be observed in individual $Ti_3C_2T_x$ nanosheets, which further reduce the screening effect of gate voltage. High-resolution TEM (HRTEM) images (Fig. 1d) represent the obvious single crystallinity structure of $Ti_3C_2T_x$ with a hexagonal symmetry arrangement of atoms and a lattice space of 0.27 nm which is further demonstrated by selective area electron diffraction pattern (Supplementary Fig. 2)[28,30].

The distribution and the thickness of $Ti_3C_2T_x$ flakes were further investigated by optical microscope and atomic force microscopy (AFM). As illustrated in Fig. 1e, the individual nanosheet is in the form of a layered structure with various sizes and shapes and the MXene flakes are staked to form continuous film with perforation under a concentration of 3 mg/ml. The thickness of $Ti_3C_2T_x$ nanosheets is extracted to be around 1.7 nm and the roughness of MXene film is 1.81 nm based on the AFM image in Fig. 1f. To verify the formation of MXene, X-ray diffraction measurement was performed with the results shown in Supplementary Fig. 3, which exhibits a characteristic peak (002) of $Ti_3C_2T_x$ at $2\theta = 7.1$[31,32]. Figure 1g demonstrates the absorption spectrum of different films, where the $Ti_3C_2T_x$ film exhibits strong absorption of UV light and weak absorption of visible light, while the absorption of semiconductor layer PDVT-10 is in the visible light and near-infrared light.

Figure 2a shows the schematic device structure of MVOFET, where MXene, semiconductor layer and drain electrode are vertically stacked, and thus the current flows through PVDT-10 from bottom MXene to top Au electrode when a negative drain voltage is applied and the carrier transport distance is determined by the

thickness of PDVT-10. The microscope image of MVOFET is provided in Supplementary Fig. 4. Figure 2b presents the effect of the concentration of MXene on the performance of MVOFET. As shown in Fig. 2b, the on-state current and the stability of devices with MXene processed from solutions with concentration of 1 and 2 mg/ml are relatively poor. By contrast, the solution with concentration of 3 mg/ml leads to devices with the best performance. This can be further verified by the SEM images of MXene obtained from solutions with concentration of 1 and 2 mg/ml (Supplementary Fig. 5), which show the MXene is sparsely distributed, resulting in poor conductivity. Supplementary Fig. 6 further demonstrates the device performance under a higher concentration of 4 mg/ml. With the increase of concentration the shielding effect of gate electric filed was enhanced, which hindered the performance of MVOFET. Figure 2c, d presents the transfer curves of MOVFEF with a concentration of 3 mg/ml, exhibiting an on/off ratio of $2 \times 10^5$, a small threshold voltage of $-1.2\,V$ and a high on-state current density of $5.8\,mA/cm^2$. Especially, the SS of MVOFET reaches 73 mV/dec, which is close to the limitation of 60 mV/dec at room temperature (Supplementary Fig. 7a). Supplementary Fig. 8 presents the distribution histogram of basic transistor performance of 30 MVOFET devices. The bias stability of device is presented in Supplementary Fig. 9, from which we see acceptable device variation and a small right shift of threshold voltage after positive gate bias. Moreover, a saturation regime, which is essential for all transistor-based applications, is observed in the output characteristic in Fig. 2e and

Supplementary Fig. 10a. When positive $V_{DS}$ was applied, holes were injected from the Au-semiconductor interface, and therefore the Ohmic contact was formed by Au and PDVT-10 resulting in a marginal modulation of $I_{DS}$ at different $V_{GS}$ values. While the application of negative $V_{DS}$ led to hole injection from MXene-semiconductor interface, and in this case the SB formed by MXene and PDVT-10 was efficiently modulated by $V_{GS}$ and thus governed the total $I_{DS}$[33].

For comparison, the VOFET with AgNWs source electrode is also demonstrated in Fig. 2f–h. The SEM image illustrates that AgNWs are randomly stacked to form a network structure. The morphology and thickness of AgNWs are demonstrated by AFM and height profile images, respectively. The stacked AgNWs have thickness of 30 nm and width of 200 nm, indicating more rough morphologies compared with MXene nanosheets (Supplementary Fig. 11). The conductivity of MXene film and AgNWs are also compared in Supplementary Fig. 12. Benefiting from the ultra-short channel length of AgNWs-based VOFET and the relative high conductivity of AgNWs, the device exhibits a large on-state current density of $9.2\,mA/cm^2$, a switch ratio about $10^5$, a relative large SS of 350 mV/dec even though it is still smaller than most planar organic transistors (Supplementary Fig. 7b)[34,35]. Besides, as shown in Fig. 2h, the output characteristics of the VOFETs using AgNWs exhibits an absence of saturation of drain current. The device performance of graphene based VOFET is also shown in Supplementary Fig. 13, in which a large on-state current alone with large SS was observed.

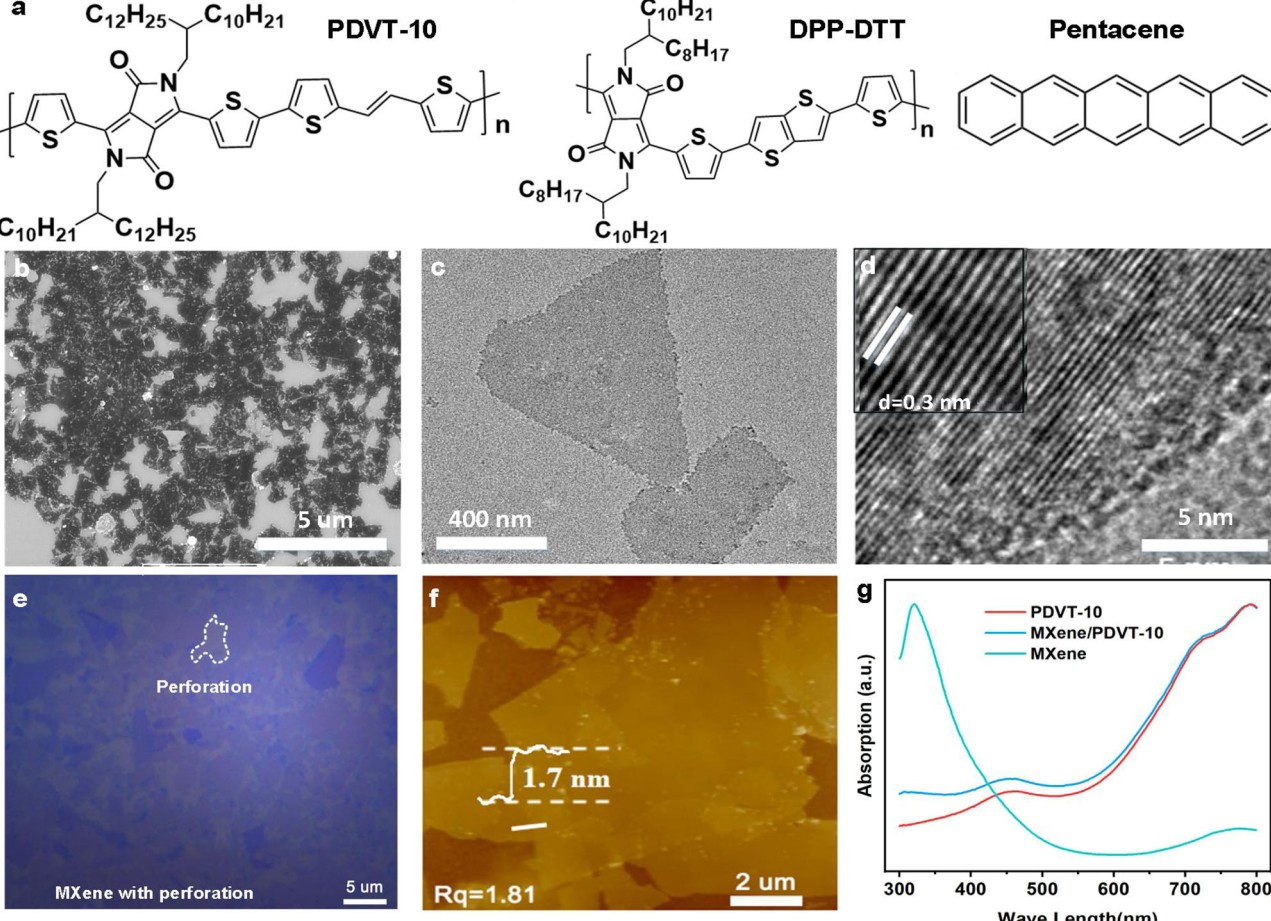

**Fig. 1 Structure of semiconductor and characteristics of MXene film. a** The chemical structure of PDVT-10, DPP-DTT and pentacene. **b** SEM image of MXene film with a concentration of 3 mg/ml. **c** TEM image of dispersive MXene nano flake. **d** HRTEM image of MXene. **e** Microscope image of MXene film with perforation. **f** The AFM image of staked MXene. **g** The UV–vis absorption spectrum of MXene and PDVT-10.

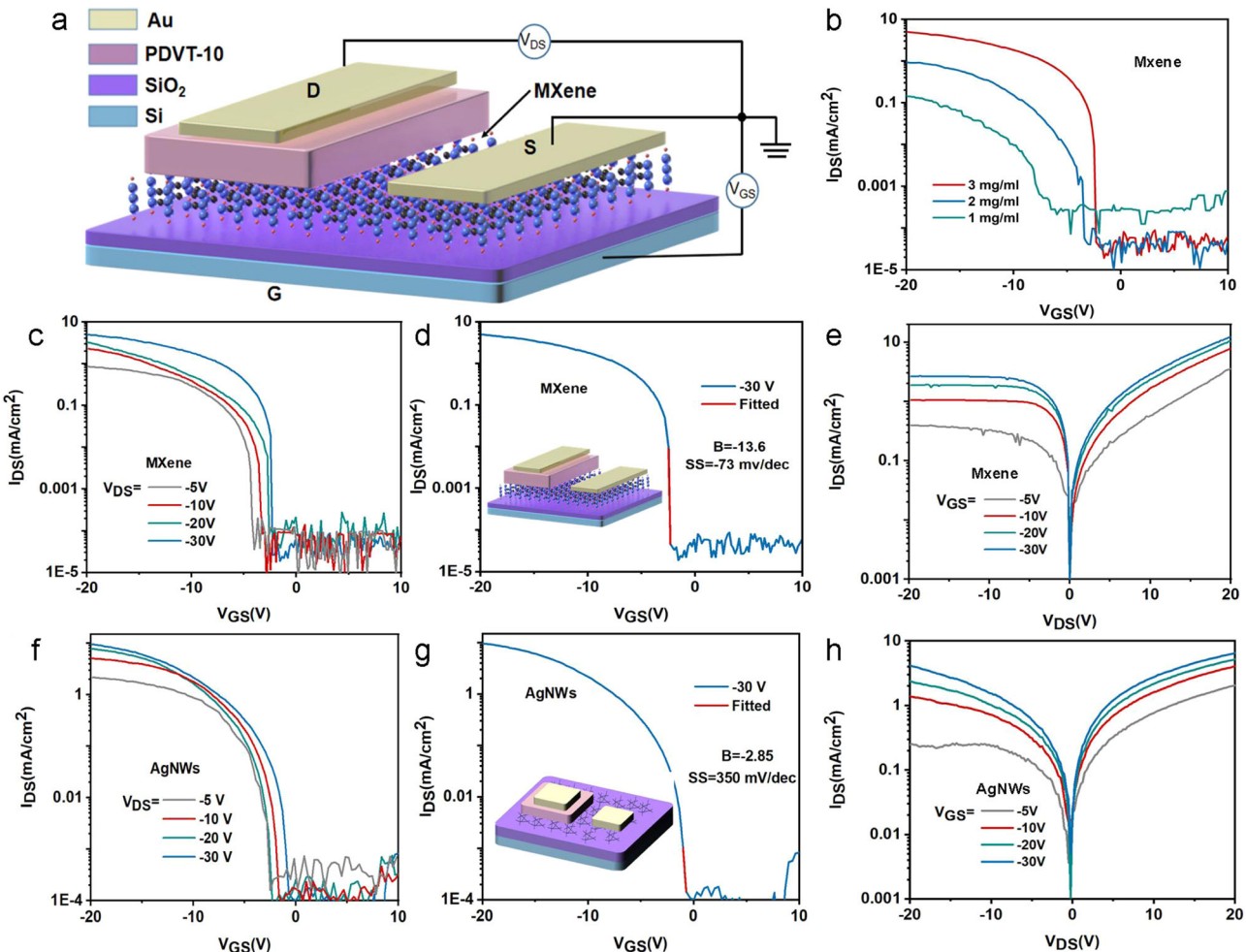

**Fig. 2 The essential transistor performance of MVOFET. a** The schematic diagram of device structure of MVOFET. **b** The transfer curves of MVOFET under different concentration of MXene. **c** The transfer curves of MVOFET under different $V_{DS}$ at a concentration of 3 mg/ml MXene. **d** The transfer curve at $V_{DS} = -30$ V, indicating a high on/off ratio over $10^5$, small threshold voltage of $-1.2$ V and small SS of 73 mv/dec. **e** Output characteristic of MVOFET exhibiting a significant saturation current under small $V_{DS}$. **f** The transfer curves of VOFET utilizing AgNWs as source electrode. **g** The transfer curve at $V_{DS} = -30$ V of VOFET made by AgNWs. **h** Output characteristic of AgNWs-based VOFET under different $V_{GS}$.

To further demonstrate the superiority and university of MXene based VOFET, the performance of two other typical semiconductors DPP-DTT and pentacene are shown in Supplementary Figs. 14 and 15. The comparison of the transistor performance of MVOFETs with other reported organic vertical transistors is summarized in Table S1, which shows the superior property of MXene as the source electrode of vertical transistors.

In order to explore the origin of the excellent gate control ability of MVOFET with a low SS, the charge injection at the MXene and AgNWs/PDVT-10 interfaces are quantitatively illustrated (Fig. 3a and Supplementary Fig. 16) through the test of temperature-dependent output characteristics. The Schottky current can be described utilizing thermionic emission model following the function as[36]:

$$I_S = AA^*T^2\exp^{-\frac{q\varphi_B}{k_BT}} \qquad (1)$$

in which $k_B$, $T$, and $q$ are the Boltzmann constant, absolute temperature, and elementary charge, respectively, and $\varphi_B$ is the SB height at the MXene/PDVT-10 interface; $A$ is the area of Schottky contact; $A^*$ is the effective Richardson constant. Therefore, the SB height under different gate voltages can be calculated from the slope, which is extracted by the point of $\ln(I_S/T^2)$ vs. $q/k_BT$ of each curve (Fig. 3b). As shown in Fig. 3c, the SB decreases as the gate voltage is increased, with SB height lowered from 0.39 eV at

Vg $= -10$ V to 0.21 eV at Vg $= -30$ V, indicating the efficient modulation of gate bias on the SB height. The transfer curves of MVOFET as a function of temperature varying from 313 to 353 K are investigated, as shown in Supplementary Fig. 17, which indicates that the channel current increases with the increase of temperature. Figure 3d–f demonstrates the temperature-dependent output characteristics of AgNWs-based VOFET and the equivalent SB height as a function of gate voltage. As shown in Fig. 3f, the injection barrier of AgNWs-based VOFET is higher and the variation of barrier height with varying Vg from $-30$ V to $-10$ V is smaller than that of MVOFET, suggesting a stronger gate voltage tunability in MVOFET, which is believed to account for the smaller SS of MVOFET.

Moreover, the MXene film possesses an ultra-thin thickness of about 1 nm which is comparable to the thickness of a single layer graphene. Therefore, we suspect that the gate electric field can also pass through the ultra-thin MXene film to further regulate the SB between MXene and semiconductor. To verify this assumption, a continuous monolayer MXene film without perforation is constructed utilizing a lateral self-assembly method[37]. Fig. 3g is the microscope image of stacked MXene flakes without perforation. Fig. 3h shows ten cycles of transfer curves of MVOFET based on the stacked single layer MXene flakes without perforation and Fig. 3i indicates the bias stress ability of the

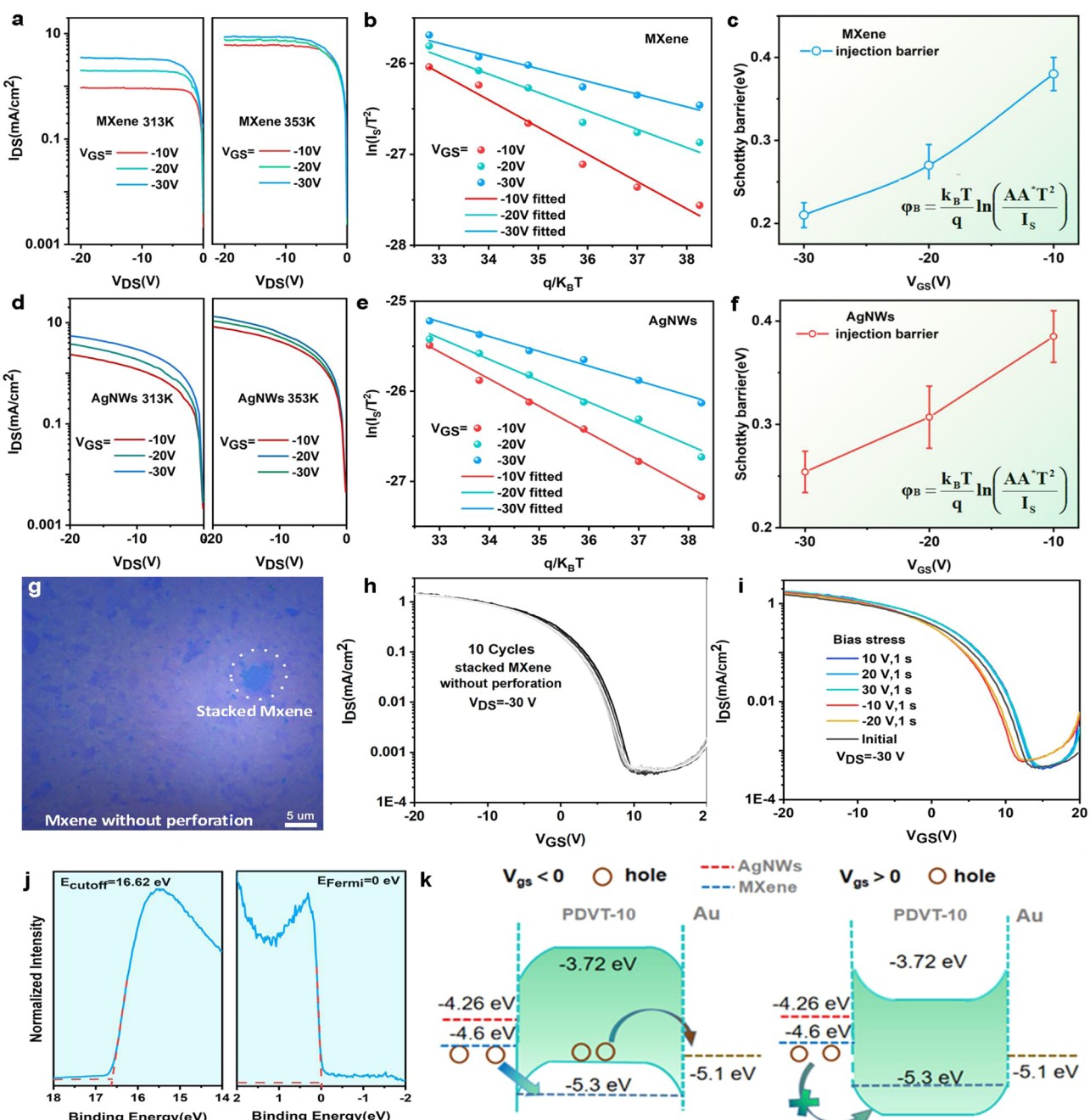

**Fig. 3 The working mechanism of MVOFET and AgNWs-based VOFET. a** The output characteristic of MVOFET under 313 and 353 K. **b** $\ln(I_{SA}T/T^2)$ vs. $q/K_BT$ plots at various gate voltages. **c** The variation of Schottky height under different gate voltage. **d** The output characteristic of AgNWs-based VOFET under 313 and 353 K. **e** $\ln(I_{SA}T/T^2)$ vs. $q/K_BT$ plots at various gate voltages of AgNWs-based VOFET. **f** The variation of Schottky height under different gate voltage of AgNWs-based VOFET. **g** The microscope image of stacked MXene flakes without perforation. **h** Ten cycles of transfer curves of MVOFET without perforation. **i** The bias stress ability of the device, **j** UPS measurement of MXene film. **k** Energy-band diagram of MVOFET and AgNWs-based VOFET. The error bars in **c**, **f** represent the standard deviation which are derived from five MVOFETs and five AgNWs-based VOFETs.

device. Compared with the device construct by stacked MXene with perforation, the device without perforation exhibits larger off-state current, relative poor gate control ability (large SS), and improvement of device stability. The device exhibits typical p-type transistor curves with an on/off ratio over $10^3$, indicating that the channel current can be regulated by gate voltage even without perforation. These results show that the gate electric field is just partially screened by the continuous MXene film, while the residual gate electric field penetrates MXene and modulates SB, which is similar to single layer graphene. The relative poor SS is ascribed to the shielding of gate electric filed. The device stability

of MVOFET with perforation is relative poor than the device without perforation, which can be attributed to the accumulation and depletion of carriers in the perforation, resulting in the shift of threshold voltage. Therefore, the MVOFET with perforation can utilize two modes (graphene and mesh metal electrode) in a single material system, which significantly improves the gate control capability of the device. Besides, the contact interface between MXene and semiconductor is significantly better than that with AgNWs. Compared with AgNWs (11.7 nm), the roughness of MXene film is reduced to 1.81 nm, which significantly reduces the contact resistance of MXene and PDVT-10.

The interface trap density of MVOFET is also compared through the double sweep transfer curves. As shown in Supplementary Fig. 18, the variation of threshold voltage of MVOFET is two times smaller than that of AgNWs-based VOFET. Therefore, MXene provides excellent interface with semiconductor layer, which further reduces the SS of MVOFET.

To demonstrate the working mechanism of MVOFET, the work function of MXene was quantitatively measured utilizing ultraviolet photoelectron spectroscopy (UPS, Fig. 3j and Supplementary Fig. 19). As shown in Fig. 3j, a visible broadened step-like feature can be observed in the spectroscopy under binding energy of 0 eV, indicating the metallic characteristic of MXene and the work function is extracted to be 4.6 eV. The schematic diagrams showing the energy level between PDVT-10 and source/drain electrodes are shown in Fig. 3k. It is seen the SB height of MXene/PDVT-10 contact is smaller than that of the AgNWs/PDVT contact, which is consistent with the equivalent SB height measured in Fig. 3c, f, resulting in the small contact resistance between MXene and PDVT-10. The working mechanism of MVOFET can be described as follows: the holes are accumulated at the interface between semiconductor and insulator layer in the openings of source electrode under negative gate voltage, resulting in energy level bending of PDVT-10 and subsequent reduction of equivalent SB height. Therefore, carriers can tunnel from the source electrode to the semiconductor when drain voltage is applied, leading to the on-state current of device. In contrast, when positive gate voltage is applied, the SB height gets increased, and in this case charge carriers cannot cross the barrier, resulting in the off-state current of device.

After the analysis of small SS of MVOFET, we now discuss the output current of device. The absence of saturation current of AgNWs-based VOFET is accounted by the leakage current between the source and brain electrode, which originates from the large roughness of AgNWs and poor contact with semiconductor. In this case, the distance between source and drain electrode decreases, resulting in large electric field between the source and drain electrodes. Therefore, the drain current increases with the increase of $V_{DS}$, resulting in the unsaturation of output current. Further evidence is demonstrated by simulation with COMSOL Multiphysics as shown in Fig. 4, which illustrates the potential and current distribution of VOFET with AgNWs and MXene as source electrode. The detailed information of theoretical modeling is described in supporting information. Fig. 4a is the schematic of the simulated device structure, which is the cross section of vertical transistor at the perforation of source electrode. It is generally assumed that the current of VOFET consists of two parts, the leakage current between source and drain electrode and the channel current in the perforation of source electrode (Fig. 4b)[17,38]. Generally, the holes are accumulated at the interface formed by the insulator and semiconductor layer of the perforation area under negative gate voltage and the charge flows through the semiconductor to the drain electrode when the drain voltage is applied (Fig. 4c). Figure 4d, e shows the potential distribution of VOFET device with AgNWs (20 nm thickness and 100 nm length) and MXenes (1 nm thickness and 100 nm length). Fig. 4g, h is the 1D potential distribution extracted from the red dashed line in Fig. 4d, e, from which we observe that the electric field between source and drain electrode in AgNWs device is larger than that in MXene device. The large electric field can induce leakage current between source and drain electrode, resulting in the absence of saturation of output current. Meanwhile, the variation of electric field in the middle of perforation is negligible, indicating that the accumulation of carriers there is not influenced by the thickness of electrode, which is consistent with reported work[39]. Moreover, the MXene flake can be spontaneously oxidized under oxygen environment, resulting in the

formation of TiO$_2$ component[40,41], which can effectively lower the leakage current between Mxene surface and the drain electrode. To illustrate the influence of native oxidation of MXene on the device performance, Fig. 4f demonstrates the potential distribution of device when 2 nm TiO$_2$ is added on MXene layer. Fig. 4i is the normalized current distribution extracted from the red line in Fig. 4f. As shown in Fig. 4f, the injection of carriers only occurs at the edge of source electrode, while the carriers are prevented from transporting to the drain electrode due to the incorporation of 2 nm TiO$_2$ layer over the top facet of MXene, eliminating the leakage current between source and drain electrode. Consequently, the conductive channel is only formed at the perforation, which contributes to the saturation of output current. To ensure the consistency of simulation and experiment, TiO$_2$ layers with different thickness (1, 2 and 4 nm) were deposited above MXene by atomic layer deposition (Supplementary Fig. 20). After the deposition of TiO$_2$ film, the devices exhibit typical p-type transistor performance and saturation of output current can be observed even with 1 nm TiO$_2$. Meanwhile, the current density decreased with a right shift of threshold voltage when the thickness of TiO$_2$ is increased. Therefore, the saturation mechanism of the output current of MVOFET is proved from the perspective of theoretical simulation and experiments.

To verify the existence of TiO$_2$ composites in Ti$_3$C$_2$T$_x$, the chemical natures of the elements were investigated by X-ray photoelectron spectroscopy (XPS) in Supplementary Fig. 21. The appearance of peaks at 458.5 eV corresponding to TiO$_2$ and the relative small intensity compared with the peak at 454.7 eV suggested that partial oxidation occurred on the surface of MXene nano-flakes[42,43]. Such combination of MXenes with TiO$_2$ nano-particles has great potential for photodetection applications. To demonstrate the photodetection performance of MVOFET, the device was illuminated with different light intensity and wavelength (Fig. 5). Fig. 5a illustrates the schematic diagram of the device under light illumination and Fig. 5b shows the transfer curves of MVOFET irradiated by different wavelengths of light under $V_{DS} = -30$ V. As shown in Fig. 5b, the MVOFET responses to UV light faster than to visible light, demonstrating the feasibility of wide-spectrum detection. Therefore, the photodetection performance of MVOFET under UV light is systemically investigated. Fig. 5c shows the transfer curves of MVOFET obtained by varying incident light intensity from 10 uW to 500 uW at $V_{DS} = -30$ V under 365 nm illumination. Compared to transfer characteristic measured in the dark condition, the $I_{DS}$ measured at $V_{GS} = 0$ V increased with the increase of light intensity and a significant right shift of transfer curves is observed. In order to better evaluate the light response performance of MVOFET, the responsivity ($R$), a parameter reflecting the photoelectric conversion ability is evaluated based on the following formula:

$$R = \frac{I_{\text{light}} - I_{\text{dark}}}{AP_{\text{in}}} \qquad (2)$$

where $I_{\text{light}}$ and $I_{\text{dark}}$ are the $I_{DS}$ with and without illumination, respectively. $A$ is the effective device area and $P_{\text{in}}$ is the incident light power. As shown in Fig. 5d, the $R$ of device exhibits the same trend with transfer curves, indicating that the $R$ parameter can be modulated by gate voltage and the highest $R$ (366 A/W) is obtained under weakest light intensity of 10 uW/cm$^2$ at $V_{GS} = -20$ V. Another important parameter, photosensitivity ($P$) can be calculated with equation: $P = (I_{\text{light}} - I_{\text{dark}})/I_{\text{dark}}$. As shown in Fig. 5e, the $P$ under positive gate voltage is much larger than the one under negative gate voltage and the $P$ increases with the increase of light intensity due to the increase of photo-induced carriers and reaches $7 \times 10^4$ under illumination intensity of

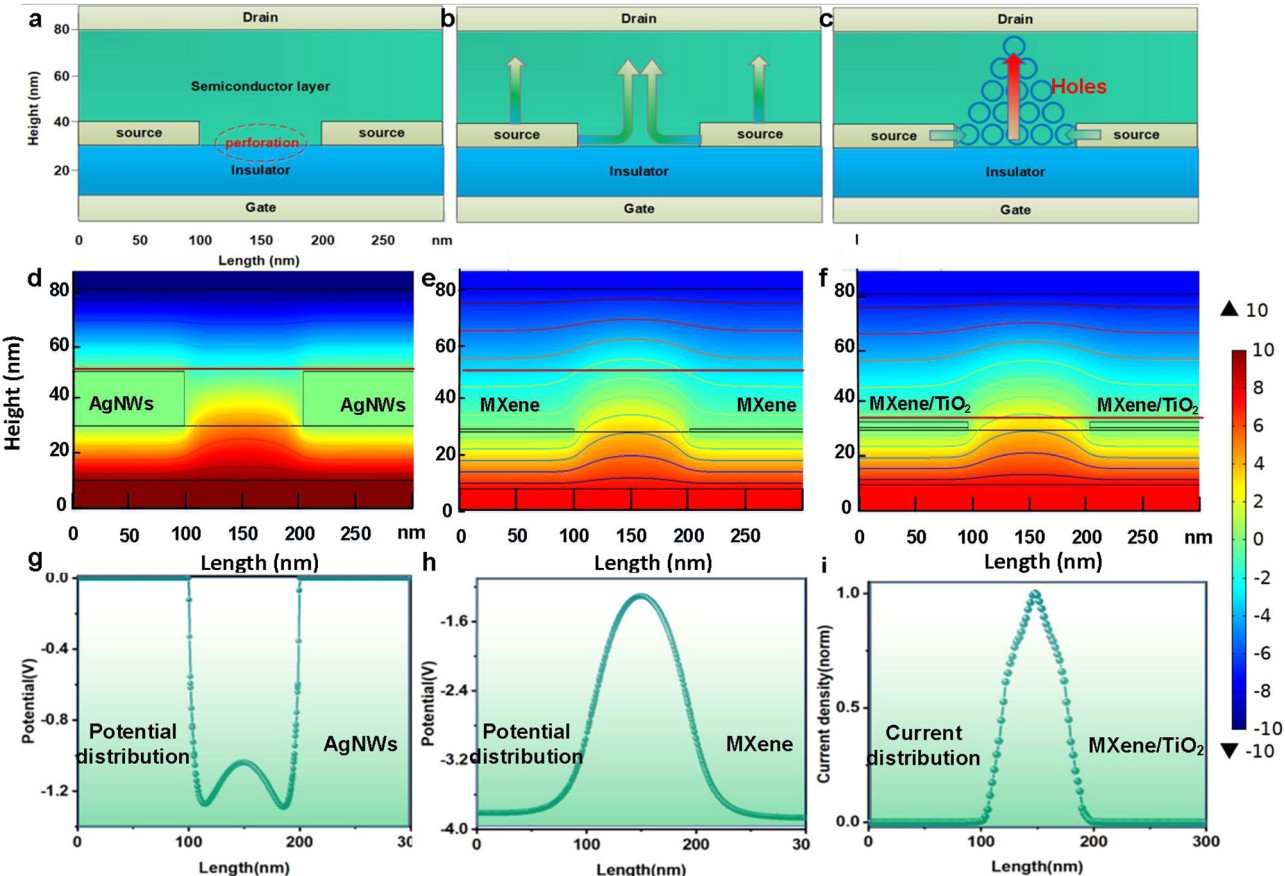

**Fig. 4 The potential and current distribution of VOFET as a function of the geometry of source electrode. a** The schematic of simulated device structure. **b** The current distribution of VOFET. **c** The schematic of the accumulation of holes and the formation of pseudo-conductive channel. **d, e** The potential distribution of AgNWs and MXene based VOFET under $V_{GS} = -10$ V, $V_{DS} = -10$ V, respectively. **f** The potential distribution of MXene based VOFET covered with 2 nm TiO$_2$. **g, h** The 1D potential distribution of the red dashed line in **d, e. i** The 1D current distribution of MVOFET extracted from the red dash line in **f**.

500 uW/cm². In addition to photosensitivity and responsivity, the detectivity ($D$) is another significant figure of merit, which evaluates the detection ability to the weak signal. The $D$ is defined with following equations:

$$D = \frac{\sqrt{A\Delta f}}{\text{NEP}} \qquad (3)$$

$$\text{NEP} = \frac{\sqrt{i_n^2}}{R} \qquad (4)$$

where $\Delta f$ is the bandwidth, $i_n$ is the noise current and NEP stands for the noise equivalent power. In general, when only shot noise from dark current is considered, $D$ can be simplified as[10]:

$$D = \frac{R}{\sqrt{2qJ_{\text{dark}}}} \qquad (5)$$

where $q$ is the elementary charge ($1.6 \times 10^{-19}$). Consequently, the maximum detectivity of MVOFET is calculated to be $2.8 \times 10^{12}$ Jones under a light intensity of 10 uW/cm² (Supplementary Fig. 22). Fig. 5f summarizes the variation of $R$ and $D$ as a function of incident light intensity, indicating that both $R$ and $D$ were negatively correlated with light intensity and reached maximum values at 10 uW/cm².

Furthermore, the transient photoresponse behaviors, which reflects the speed of photoresponse of device were further investigated. In order to explore the speed of photoresponse of MVOFET, the real-time response of device with light turned on and off was conducted at bias of $V_{GS} = 10$ V and $V_{DS} = -20$ V.

Figu. 5g shows the cyclic stability of optical response of device under UV illumination (20 cycles), in which the current under illumination is defined as "1" state and the dark current was defined as "0" state. As shown in Fig. 5g, the current-time response rose rapidly when UV light was turned on and quickly returned to the initial state when illumination was turned off. Moreover, it can be seen the MVOFET device has highly reproducible and reversible photoresponse and excellent stability. To further illustrate the rising and falling time with and without illumination, a single temporal light response is extracted from Fig. 5h. As shown in Fig. 5h, the rising and falling time is calculated to be 10.4 and 14.2 ms, respectively, which are significantly faster than that of planar phototransistors and other VOFET based photodetectors[10,44,45]. It is notable that the response speed of MVOFET is limited by the resolution of test instrument (10 ms), and thus the switching speed shown here can be potentially higher. To make a comparison of the response of MOVFET under UV and visible light, the transient photoresponse behaviors under 500 and 700 nm are investigated and presented in Supplementary Fig. 23. Table S2 summarizes the comparison of photodetection performance between MVOFET and other reported works. As shown in the table, the performance of MVOFET is comparable with that reported in literatures. Particularly, the response speed is among the highest levels. Generally, a complex layer or bulk heterojunction structure is required for broad-spectrum detection. Here the MXene-TiO$_2$ composites induced by the native oxidation of MXene broadens

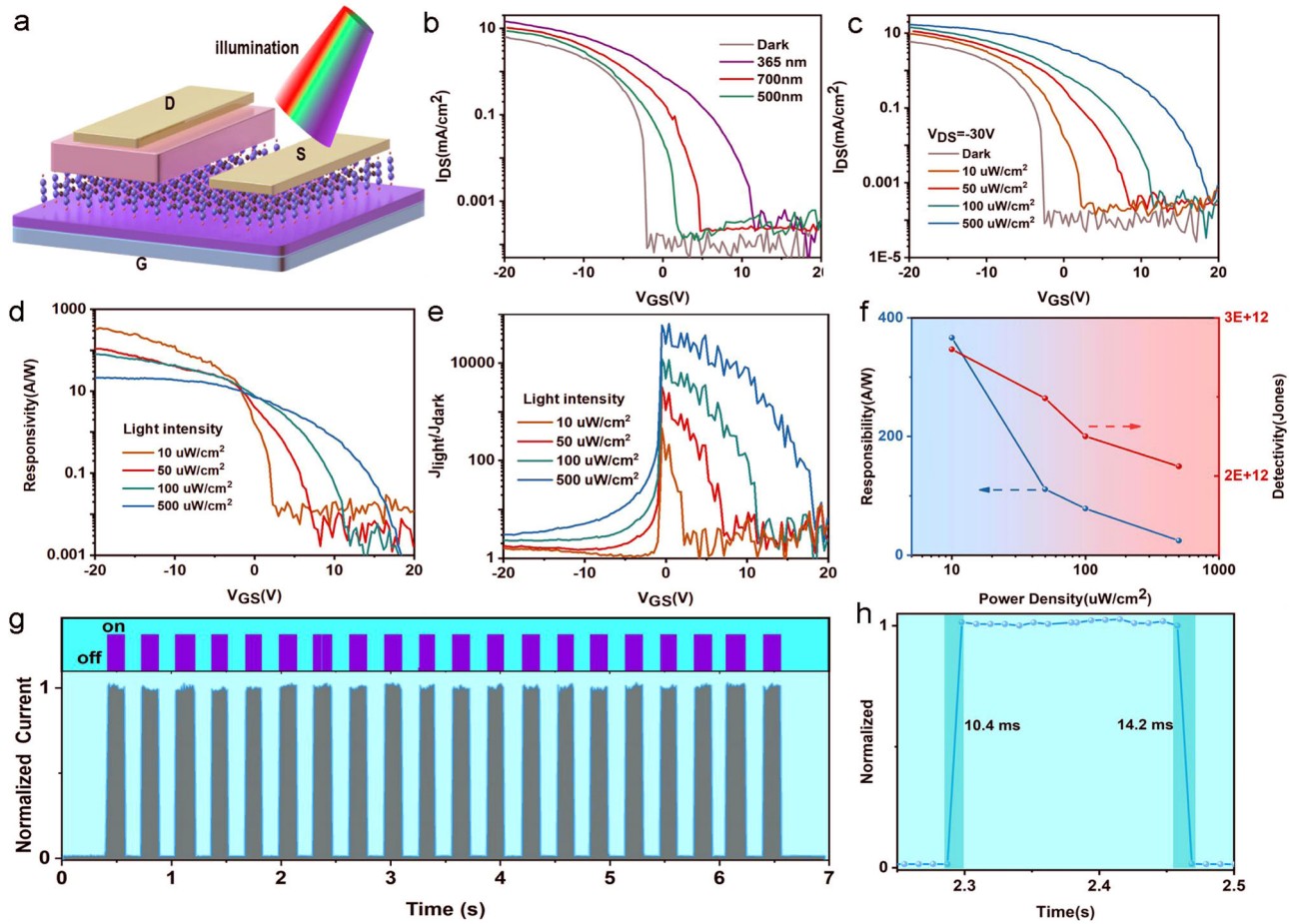

**Fig. 5 The optoelectronic performance of MVOFET. a** The schematic of MVOFET under illumination. **b** Transfer curves of MVOFET under dark and different wavelength incident light. **c** The transfer curves measured at different light intensity under 365 nm. **d, e** Responsivity and Photosensitivity as a function of gate voltage under different light intensity measured at 365 nm, respectively. **f** Responsivity and specific Detectivity under different light intensity at 365 nm. **g** The real-time UV light response cycle test measured at $V_{GS} = 10$ V and $V_{DS} = -20$ V. **h** A single light response under 365 nm, exhibiting a fast rising and falling time of 10.4 and 14.2 ms, respectively.

the absorption spectrum of device with no need of additional materials and complex device structure design. The schematic diagram of mechanism under UV and visible light is illustrated in Supplementary Fig. 24. The faster response speed to UV light than to visible light is ascribed to the stronger light response of TiO$_2$ nanoparticles and the larger HOMO level difference between TiO$_2$ and PDVT-10, resulting in the faster transfer of photo-generated holes to PDVT-10 layer.

Finally, the superiority of MVOFET compared with other VOFET is concluded. (1) The solution processable property of MXene endows MVOFET with the feasibility of large-scale fabrication and the adjustable work function (terminal ligand engineering) allows MVOFET to be more compatible with organic semiconductors. (2) The combination of both SB modulation modes of graphene and mesh metal electrode in a single material system and the excellent contact between MXene and organic semiconductor significantly improves the SS of MVOFET. (3) The issue of unsaturation of output current in VOFET is solved by using MXene as source electrode due to its ultra-thin thickness and native oxidation. (4) The native oxidation of MXene broadens the detection spectrum of VOFET without additional layer and the ultra-thin thickness offer great prospects in ultra-thin and 2D semiconductor based VOFET.

## Discussion

In summary, we demonstrate an organic vertical photoelectric transistor with low SS and saturation characteristic by utilizing

2D MXene material as the source electrode. Benefiting from the nanoscale channel length, the improvement of gate control ability and the excellent contact between 2D MXene and semiconductor, MVOFET exhibits a small SS of 73 mv/dec with threshold voltage of −1.2 V. Besides, the saturation of output characteristic is easily achieved in MVOFET due to the ultra-thin thickness of MXene film and native oxidation of MXene, which together eliminate the leakage current between source and drain electrode. Moreover, MVOFET possesses great potential in wide-spectrum photo-detector with fast response speed under UV (10 ms) and visible light (0.21 s) without additional material and structure design. Hence, MVOFET shows great potentials for high-performance VOFETs and related optoelectronic devices.

## Methods

**Materials.** High molecular weight π-extended organic p-type semiconductor copolymer poly[2, 5-bis(alkyl)pyrrolo-[3,4-c]pyrrole-1,4(2H, 5H)-dione-alt-5, 50-di(thiophen-2-yl)-2, 20-(E)-2-(2-(thiophen-2-yl)vinyl)thiophene] (PDVT-10) (Mw = 183 K) and poly(N-alkyl-diketopyrrolo-pyrroledithienylthieno [3, 2-b] thiophene) (DPP-DTT) were obtained from 1-Materials[46]. PDVT-10 and DPP-DTT were dissolved in chloroform with concentration of 5 mg/ml for the preparation of semiconductor layer and pentacene was deposited by thermal evaporation. MXene (5 mg ml$^{-1}$ in deionized water) was purchased from XFnano Materials Tech Co., Ltd. and further diluted to 4, 3, 2 and 1 mg/ml as the source electrode of device. The AgNWs (5 mg/ml in isopropanol) was purchased from Suzhou ColdStones Technology Co., Ltd and diluted to 0.5 mg ml$^{-1}$ using isopropanol for preparation.

**Device fabrication.** The diluted MXene solution was deposited on cleaned Si wafer with 100 nm SiO2 by spin coating at 1000 rpm for 10 s followed by 2000 rpm for 30 s. The AgNWs was also deposited by spin coating at 2000 rpm for 60 s. After the deposition of source electrode, a 50 nm gold source was thermally evaporated onto the AgNws through shadow mask to serve as contact electrode. After that, the PDVT-10 film was coated on the substrate by spin coating at 1000 rpm for 60 s and then annealed at 150 °C for 10 min. To pattern the semiconductor layer, the sample was immersed in chloroform solvent to remove the excess PDVT-10 so that the deposited gold electrode will be completely exposed. Finally, a 50 nm gold drain electrode was deposited through thermal evaporation on the top of remaining PDVT-10 to serve as drain electrode.

**Device characterization.** The electrical characteristics and photodetector performance were measured by semiconductor parameter analyzer (Keysight 2912) in ambient conditions. The surface morphology of MXene and AgNWs was measured by AFM (Bruker MultiMode 8). The SEM images of MXene were obtained on a focusion beam/SEM (Nova NanoSEM 230). The TEM was performed with a JEOL JEM-2100 microscope operated at 200 kV. The HRTEM images were acquired by a Gatan CCD camera operated in electron-counting mode. The UV–vis absorption spectra for films were tested on quartz substrates with Cary 300 UV–vis spectrometer (AgilentTechnologies). The UPS and XPS were conducted at constant analyzer energy model with energy step size of 0.02 and 0.1 eV, respectively (Thermo Scientific ESCALAB 250). The monochromatic light was produced by a 300 W wavelength-adjustable xenon lamp source (Beijing NBET Technology Co., Ltd., Omno302).

## Data availability

The data that support the findings of this study are available from the corresponding author upon reasonable request.

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

## Acknowledgements

The authors are grateful for financial support from National Natural Science Foundation of China (U21A20497, 61974029), the Natural Science Foundation of Fujian Province (2020J06012, 2020J05104) and the Fujian Science and Technology Innovation Laboratory for Optoelectronic Information of China (2021ZZ129).

## Author contributions

H.P.C. and T.G. conceived the project, E.L. designed and performed the experiments and collected the data. E.L., C.G., R.Y., X.W., L.H., Y.H. and H.J.C. analyzed and discussed the data. H.P.C. supervised the project. E.L. wrote the paper.

## Competing interests

The authors declare no competing interests.
