## [Peer Review File · Nature Communications]

Title: MXene based Saturation Organic Vertical Photoelectric Transistors with Low Subthreshold SwingReviewers' comments:

Reviewer #1 (Remarks to the Author):

Comments on manuscript: MXetronics based Saturation Organic Vertical Photoelectric Transistors with Low Subthreshold Swing (NCOMMS-21-34711)

The authors use partially oxidized Ti₃C₂T_x MXene as the source contact in an organic vertical transistor and demonstrate the low subthreshold swing in the transistor transfer curve. The photoresponse of the MXene contact transistor is also reported.

This reviewer cannot recommend this article for publication in Nature Communications for the following reasons.

1. The authors don't really provide a good mechanistic explanation of why MXenes have improved the SS. This is critical information that is conspicuously missing from the manuscript.

2. The authors touch on this issue in very general terms saying: "unlike the rod-shaped AgNWs, the 2D flake structure of MXene not only introduces a high quality interface with a roughness of 1.47 nm but also increases the contact area with semiconductor".

What is exactly a "high-quality" interface? Have the authors characterized interface trap density? Have the authors looked at bias stability? What is the nature of the bonding between MXene surface charges and the semiconductor? Are there dangling bonds?

Moreover, the MXene film roughness of about 1.47 nm measured by AFM image is substantially smaller than the device area. From the SEM image, the MXene films appear more rough. So at the device scale, the MXene roughness appears higher. So neither the "interface quality" nor the roughness argument are convincing to explanation.

For Nature Communications, more than just fabricating the device and measuring its characteristics needs to be done.

3. The device, while having low SS, has poor Ion/Ioff ratio, and there is no discussion why the MXene did not improve the Ion/Ioff or mobility. An improved interface should improve all these parameters.

4. The authors claim that "flake-like geometry provides excellent contact and tunable barrier height". How can flake-like geometry provide tunable barrier height? This seems like new physics to me.

5. The band diagram in Figure 3h is confusing, V_g can adjust the channel layer work function and thus the barrier height between MXene and VBM is adjusted accordingly. Why do the authors give a fixed VBM of channel layer at -5.3 eV for both positive V_g and negative V_g operation?

6. In Figure 2a, the authors only give device schematic, but no real images of the devices. The authors should provide real device images and give more details on the device size, such as channel layer thickness, contact pad size, the distance between contacts, and MXene layer conductivity.

7. In the introduction part and later in the main text, the authors made several claims about the leakage current effect in the Vertical transistor, but the authors did not provide any experimental data on leakage currents. Since the leakage current is addressed as one of the key parameters, authors should give more analysis on it.

8. From Fig. 1g, it is hard to say that MXene/PDVT-10 bilayer film has enhanced absorption in UV light based on such a slight difference.

9. On page 8, Authors only tried MXene solution with 1, 2, 3 mg/ml and then claimed 3 mg/ml is the best. Any idea what happens at higher concentration? Is there a point where the concentration becomes too high to get a good performance?

10. On page 9, the authors claim that "SEM and AFM images of AgNWs deposited on SiO₂ substrate with a concentration of 0.5 mg/ml (optimized concentration)" Any data to support this optimization?

11. In Figure 3g, the authors only give the UPS data of MXene. Why not also give the AgNWs UPS data?

12. On page 12, the authors claim that "The superior conductivity and small sheet resistance endowed MXene unparalleled carrier transport speed." The authors should show the experimental evidence the MXene has superior conductivity than Ag NW.

13. Simulation in Figure 4 is the kind of planer transistor with symmetrical MXene or Ag NW contact in both source and drain. However, the paper is focused on vertical transistors with asymmetrical MXene contact only in the source area. Please comment on this.

14. In Figure 5, the semiconducting polymer and MXene are covered by opaque gold electrodes, so how can they detect light?

15. What about the hysteresis in these devices? Have the authors studied how the stability of the devices with respect to hysteresis and threshold voltage shift with the bias? This is important information should be carefully analyzed.

16. The title use of the term "MXetronics" here is not really appropriate. Please remove it. A more appropriate title maybe "MXene Based Saturation Organic Vertical Photoelectric Transistors with Low Subthreshold Swing"

17. There are few claims that the authors make which don't have any supporting data or seem like an

opinion:

a. The authors mentioned graphene is complicated because of CVD growth. Actually, there are many reported solution-processed graphene.

b. The authors claimed that AgNWs are unfavorable for achieving high-performance vertical transistors due to the large sheet resistance in the introduction part. AgNWs are very conductive and maybe even better than unoxidized MXene { Nano Futures 4 035002,(2020) Composites Part A: Applied Science and Manufacturing 149, 106545(2021) }..

c. the authors claim that “there are very few reports of the application of MXene in organic transistors” There are already several papers {Adv. Mater. 2021, 33, 2008215; ACS Appl. Mater. Interfaces 2020, 12, 29, 32970–32978; ACS Nano 2019, 13, 10, 11392–11400; Microchimica Acta volume 188, Article number: 301 (2021) Nanoscale, 2018,10, 5191-5197}. Again, the investigation in the background is not enough.

In short, this paper is more appropriate for a device journal because it lacks the scientific insight and the serious analysis required for publishing in Nature Comm

Reviewer #2 (Remarks to the Author):

This paper introduces the transition-metal carbides/nitrides (MXene) material as a new source electrode of organic vertical transistors, which improves the subthreshold swing (SS) to 73 mV/dec comparing with the 350 mV/dec of traditional AgNWs based vertical transistors. This application also takes advantage from the ultra-thin thickness and oxidation properties of MXene and greatly improves the saturation of output current. The experiment is explained in detail and finite element modeling cases are conducted in COMSOL Multiphysics. The modeling results support and validate their theory. This innovative application is promising and expands the application of MXene materials in the fields of electronic devices. However, there are some major problems or flaws to be addressed before being further consideration for publication.

1, In the introduction section, the electro-chemical and mechanical property backgrounds of MXene material and why it is specifically suitable for serving such purposes are not well established. This lack of explanation fails to illustrate the necessity of using this class of material rather than other 2D materials.

2, Throughout this paper, no molecular dynamics or density functional theory models are being conducted to explore the electrical and mechanical properties of the MXene in the electrode application, which would provide insights to the nanoscale. The finite element modeling using COMSOL Multiphysics alone is not enough to serve as theoretical background due to the scale of the experiment.

3, Too few samples are discussed in this work, and this reduces the generosity of the application. More experimental results are recommended to provide a statistic view of the essential properties of MXene VOFET.

4, The grammar and the spelling are recommended to be checked thoroughly to eliminate existing typos. Short straight sentences are recommended to replace the long complicate ones to help with the understanding.

Based on the reviews, I do not recommend the paper to be accepted with the current form.

Reviewer #3 (Remarks to the Author):

The authors have used MXene as the electrode for the vertical photoelectric transistors, showing a low subthreshold swing value of 73 mV/dec. The whole content may be suitable for publication in this journal if the authors can provide proper answers and corresponding revision to comments listed below,

1. It is unfair to compare the MXene 2D materials with silver nanowire with a high surface roughness that will degrade the device performance. It is predictable that the subthreshold swing of MXene based devices will be lower than that of AgNW based devices. The reviewer strongly recommends the authors compare MXene with graphene with similar conductivity to show the superior property of MXene itself.
2. In Supplementary Fig. 1, due to the large particles, the surface roughness seems much larger than 1.47 nm as described in line 151.
3. How is it possible to obtain an MXene film of a d-spacing of 1.25 nm without any post-treatment? Please refer to the following paper. Science 2020, 369, 446.
4. The distribution of TiO₂ is non-uniformly MXene as a particle instead of a film. Hence the reproducibility might be below. Please show a histogram of the device's performance to show reproducibility.
5. Please show the change in device performance with the variation of TiO₂ particle density.
6. Please use solution-processed or sputter TiO₂ layer with different thicknesses as a comparison,
7. Device simulation is lateral structure, but the actual device is vertical. Please use a similar device structure.
8. In addition, TiO₂ did not form a layer of 2 nm in the actual device.
9. In line 305, please explain the faster response of MVOFET to UV light than to visible light.
10. Please compare the device performance using MXene with other vertical photoelectric transistors from the previous results and summarize it in one figure to show the significance of this work.

We would like to thank all the reviewers for their critical suggestions and valuable comments. Reviewer's comments are in blue, while our responses are immediately below. We have added relevant experimental results to clarify the concerns of reviewer. Modifications to the manuscript are highlighted in the manuscript itself.

We have carefully revised the manuscript according to the reviewers' comments and the point-to-point response to the referee is also carefully executed. We admitted that the advance and novelty of our work is not enough in the early stage. To prove the superiority of the use of MXene in vertical transistors and improve the novelty of work, we have conducted a series of complementary experiments. According to the complementary experiments, **there is a new exciting discovery that the MXene take advantage of both graphene and mesh metal electrode to be furnished as the source electrode of vertical transistors** (the working mechanisms of VFET can be roughly divided into two categories. A class is the continuous single layer graphene and another is the mesh metal electrode), while in our previous version the working mechanism of our device is only explained by mesh metal one. **This is a completely new finding, which well explained the superiority of MXene compared with other electrodes for vertical transistors.** The ultra-thin film of MXene with perforation probes both partially shielding effect of graphene and the direct modulation Schottky barrier of the mesh electrode, which significantly improves the gate control ability of MVOFET resulting in the small SS. We believe that the supplementary experiments and our point-to-point answer of the questions would address the reviewers' concerns about our work.

Reviewer #1 (Remarks to the Author):

The authors use partially oxidized Ti₃C₂T_x MXene as the source contact in an organic vertical transistor and demonstrate the low subthreshold swing in the transistor transfer curve. The photoresponse of the MXene contact transistor is also reported.

This reviewer cannot recommend this article for publication in Nature Communications for the following reasons.

1. The authors don't really provide a good mechanistic explanation of why MXenes have improved the SS. This is critical information that is conspicuously missing from the manuscript.

“Author reply”: We thank reviewer for the careful and constructive comments to improve our work. To provide a deep mechanistic explanation of the improvement of SS, we have conducted a series of complementary experiments. On the basis of our experiment results, we have reorganized the manuscript and a new mechanism was proposed. The interface quality is not the major reason for the small SS of MVOFET. According to the complementary experiments, we are surprised to find that the MXene take advantage of both graphene and mesh metal source electrode. The ultra-thin film of MXene with perforation probes both partially shielding effect of graphene and the direct modulation Schottky barrier of the mesh electrode, which

significantly improve the gate control ability of MVOFET, resulting in the small SS.

Firstly, the operation of VFET relies on the gate-tunable modulation of Schottky barrier in the interface between source electrode and semiconductor and thus regulated the injection of carriers. Depending on the utilized of source electrode, the working mechanisms of VFET can be roughly divided into two categories. A class is the continuous single layer graphene and another is the mesh metal electrode. In the graphene based VFET, the graphene just partially screened the gate field, therefore, the gate field can penetrate graphene and then modulate the Schottky barrier. To reduce the shielding effect, single layer graphene is usually needed. However, the gate field is still screened to some extent hindering the gate control ability of graphene based VFET. Except for graphene, there are some VFETs using mesh metal electrode as source electrode. The gate field will be screened by conventional metal such as AgNWs. Therefore, to function those devices, the metallic source electrode is required to be perforated, thus allowing the gate field direct access to the metal-semiconductor interface, where a field induced band bending lowers the barrier to allow channel currents. The porous structure of source electrode reduces the shielding effect of the gate electric field and improves the gate control capability of the device. The MXene film possesses ultra-thin thickness of about 1 nm which is similar to the single layer graphene. Therefore, we envisage that the gate electric field can also pass through the ultra-thin MXene film to further regulator the Schottky barrier between MXene and semiconductor. To verify this assumption, we constructed a continuous monolayer MXene film without perforation utilizing a lateral self-assembly method as described in ACS Nano 2021, 15, 625–636.

Fig. 3. The working mechanism of MVOFET and AgNWs based VOFET. (a) The output characteristic of MVOFET under 313K and 353 K. (b) $\ln(I_{SAT}/T^2)$ versus $q/K_B T$ plots at various gate voltages. (c) The variation of Schottky height under different gate voltage. (d) The output characteristic of AgNWs based VOFET under 313K and 353 K. (e) $\ln(I_{SAT}/T^2)$ versus $q/K_B T$ plots at various gate voltages of AgNWs based VOFET. (f) The variation of Schottky height under different gate voltage of AgNWs based VOFET. (g) The microscope image of stacked MXene flakes without perforation. (h) 10 cycles of transfer curves of MVOFET based on the stacked single layer MXene flakes without perforation. (i) The bias stress ability of the device (j) UPS measurement of MXene film. (k) Energy-band diagram of MVOFET and AgNWs based VOFET.

Fig.3g is the microscope image of stacked MXene flakes without perforation. Fig.

3h shows the 10 cycles of transfer curves of MVOFET based on the stacked single layer MXene flakes without perforation and Fig. 3i is the bias stress ability of the device. Compared with the device construct by stacked MXene with perforation, the device without perforation exhibits larger off state current, relative poor gate control ability (large SS), and an improvement of device stability. The device exhibits typical p-type transistor curves with an on/off ratio over 10^3 , indicating that the channel current can be regulated by gate voltage even without perforation. These experimental phenomenon shows that the continuous MXene film just partially screens the gate electric field, while the residual gate electric field penetrates MXene and modulates Schottky barrier, which is similar to single layer graphene. The relative poor SS is ascribed to the shield of gate electric filed. The device stability of MVOFET with perforation is relative poor than the device without perforation, which can be attributed to the accumulation and depletion of carriers in the perforation resulting in the shift of threshold voltage. Therefore, the MVOFET with perforation can probe both modes (graphene and mesh metal electrode) in a single material system, which significantly improves the gate control capability of the device.

Secondly, compared with AgNWs, the work function of MXene is higher, which reduces the injection barrier of carriers. Besides, the variation of equivalent SB height of AgNWs based VOFET measured in Figure 3 is smaller than that of MVOFET under the same gate voltage, indicating a stronger gate tunability in MVOFET. Moreover, unlike the rod-shaped AgNWs, the 2D flake structure of MXene possesses much smaller roughness (1.81 nm of MXene and 11.7 of AgNWs under the same range). The variation of threshold voltage (ΔV_{TH}) measured from the hysteresis for double sweep of AgNWs and MVOFET is provided in Fig. S18. As shown in Fig. S18, the ΔV_{TH} of MVOFET is two times smaller than AgNWs based VOFET. Therefore, the MXene provides excellent interface with semiconductor layer, which further reduces the SS of MVOFET. The basic transistor performance of MVOFET and other reported vertical organic transistor are compared in Table S1 to show the superior property of MXene itself.

Semiconductor Materials	Source electrode	SS	Ion/Ioff	Current density	Threshold voltage(V)	Ref
C ₆₀	Grahphene	>2(V/dec) ^a	3×10^3	1×10^{-6} (A)	>50 ^a	1
PBDB-T	Grahphene	>2(V/dec) ^a	10^4	1×10^{-6} (A)	4	2
PC ₇₁ BM	Grahphene	>2(V/dec) ^a	10^5	1×10^{-5} (A)	-5	2
C ₆₀	Grahphene	5.75(V/dec)	8×10^4	3×10^{-6} (A)	-10	3
PTCDI-C ₈	Grahphene	>2(V/dec) ^a	< 10^3	12.4 (mA/cm ²)	>-20 ^a	4
Pentacene	Grahphene	>2(V/dec) ^a	< 10^3	10.5 (mA/cm ²)	>30 ^a	4
DNTT	CNT	500(mV/dec)	10^5	110 (mA/cm ²)	1	5

MoS ₂	CNT	>2(V/dec) ^a	10 ³	1×10 ⁻⁶ (A)	>-40 ^a	6
MAPbI ₃	Porous ITO	1.1(V/dec)	10 ⁴	1 (mA/cm ²)	1	7
CuPc	Pattern Au	>2(V/dec) ^a	10	0.5 (A/cm ²)	1	8
PDVT-8	AgNWs	500(mV/dec)	2×10 ⁴	6.5 (mA/cm ²)	1.5	9
PDVT-10	Graphene	1.75(V/dec)	1×10 ⁴	37.5(mA/cm ²)	8.5	This work
PDVT-10	AgNWs	350(mV/dec)	1×10 ⁵	9.2 (mA/cm ²)	0.7	This work
PDVT-10	MXene	73 (mV/dec)	2×10 ⁵	5.8 (mA/cm ²)	1.2	This work

^a estimated from the transfer curves of reported devices.

Supplementary Fig. 18. The double sweep measurement of (a) MXene and (b) AgNWs.

Thirdly, the solution processable property of MXene endows MVOFET the feasibility of large-scale fabrication and the adjustable working function (terminal ligand engineering) allows MVOFET more compatible with organic materials. The issue of the unsaturation of output current in VOFET is solved with MXene due to its ultra-thin thickness and native oxidation. Besides, the native oxidation of MXene broadens the detection spectrum of VOFET without additional layer and the ultra-thin thickness would offer great prospects in ultra-thin and 2D semiconductor based VOFET. Therefore, the utilization of MXene as the source electrode of VFET not only improves the device performance and solves the unsaturation issue of VFET but also offers great potential in organic electronics. Relative discussion has been added in the main manuscript.

2. The authors touch on this issue in very general terms saying: “unlike the rod-shaped AgNWs, the 2D flake structure of MXene not only introduces a high quality interface with a roughness of 1.47 nm but also increases the contact area with semiconductor”.

What is exactly a “high-quality” interface? Have the authors characterized interface trap density? Have the authors looked at bias stability? What is the nature of the bonding between MXene surface charges and the semiconductor? Are there dangling bonds?

“Author reply”: We thank reviewer for the valuable comments to improve our work. The ΔV_{TH} extracted from the double sweep of MVOFET and AgNWs based VOFET are provided to demonstrate the interface trap density. The ΔV_{TH} of MVOFET is two times smaller than AgNWs based VOFET, according to the formula $\frac{\Delta V_{TH}}{e} C_i$, which is usually used to calculate the interface trap density, the interface trap density of MVOFET is two times smaller than AgNWs based VOFET.

Supplementary Fig. 18. The double sweep measurement of (a) MXene and (b) AgNWs.

The bias stability is also provided in Fig. S9, the bias stability was measured under 5 V gate voltage with different bias time. The threshold voltage exhibits a small right shift under positive gate voltage, which can be ascribed to the accumulation of carriers in the perforation of MXene film.

Supplementary Fig. 9. The bias stability of MVOFET measured at 5V gate voltage with different bias time

In addition, according to the reference (J. Mater. Chem. A, 2021, 9, 5016–5025), the oxidation of MXene can generate Ti–O bonds which can effectively reduce the defects of the MXene film fabricated by spin-coating. The oxidation of MXene can be also served as a passivation layer of the device. Therefore, we conclude that the interface of MXene film is better than AgNWs. Relative discussion has been added in the main manuscript.

Moreover, the MXene film roughness of about 1.47 nm measured by AFM image is substantially smaller than the device area. From the SEM image, the MXene films appear more rough. So at the device scale, the MXene roughness appears higher. So neither the “interface quality” nor the roughness argument are convincing to explanation.

“Author reply”: We thank reviewer for the valuable comments to improve our work. The surface roughness indeed increased with the range of AFM image due to the oxidation of MXene. However, we compared the roughness of MXene film and AgNWs in the same range, the results and the conclusion are still important. Limited by the test instrument, we cannot provide a large range AFM image of MXene film, therefore we provide a AFM image with more stacked MXene flakes in the revised manuscript. As shown in the AFM image, the roughness of MXene film is 1.81 and the thickness of single layer MXene flake is 1.7 nm, compared with the AFM image of AgNWs measured in the same range, the roughness of MXene film is still much smaller than AgNWs (11.7). Besides, we have provided a microscope image of MXene film with large range, which also demonstrate the uniform deposition of MXene film, therefore, the overall roughness of MXene film will not increase too much. So we can still conclude that MXene has much better interface quality compared with Ag NWs. Besides, based on our new experiments, the interface quality

is not the major reason for the small SS of MVOFET. The MXene take advantage of both graphene and mesh metal source electrode, resulting in the strong gate control ability of device.

Fig. 1. Structure of semiconductor and characteristics of MXene film. (a) The chemical structure of PDVT-10, DPP-DTT and Pentacene. (b) SEM image of MXene film with a concentration of 3 mg/ml. (c). TEM image of dispersive MXene nano flake. (d) HRTEM image of MXene. (e) Microscope image of MXene film. (f) The AFM image of staked MXene. (g). The UV-vis absorption spectrum of MXene and PDVT-10.

3. The device, while having low SS, has poor Ion/Ioff ratio, and there is no discussion why the MXene did not improve the Ion/Ioff or mobility. An improved interface should improve all these parameters.

“Author reply”: We thank reviewer for the valuable comments to improve our work. The Ion/Ioff ratio of MVOFET is not smaller than the AgNWs based VOFET, as shown in Fig. 2, the on/off ratio of AgNWs based VOFET is 1×10^5 and the on/off

ratio of MVOFET is 2×10^5 , which is two times larger than that of AgNWs. The on state current and off state of MVOFET are all relative small than AgNWs, which can be attributed to the oxidation of MXene resulting in the decrease of conductivity of MXene. The conductivity of MXene film and 0.5 mg/ml AgNWs are compared in the revised manuscript, as shown in the image, the conductivity of MXene is smaller than

AgNWs. As for the mobility, there is no article to calculate the mobility of vertical transistor so far. Because unlike the lateral interface transport carriers in planar transistor, the carriers are vertical bulk transport in vertical transistors. Therefore, the formula calculated in planar devices is not suitable for vertical transistors.

Supplementary Fig. 12 The conductivity of (a) MXene and (b) AgNWs.

4. The authors claim that “flake-like geometry provides excellent contact and tunable barrier height”. How can flake-like geometry provide tunable barrier height? This seems like new physics to me.

“Author reply”: We are sorry for our ambiguous statement. To avoid misleading, the statement is modified in the revised manuscript. Indeed, the flake-like geometry cannot provide tunable barrier. It is the gate voltage that can regulate the Schottky barrier height between semiconductor and MXene heterojunction, which is the basic mechanism of vertical transistor. The meaning what we want to convey is that the working function of MXene can be regulated by modulating the terminal ligand, which is equivalent to regulating the Schottky barrier between semiconductor layer. Therefore, the choice of semiconductor materials is not fixed. The statement in the revised manuscript is modified.

“Meanwhile, their work function ranged from 2.14 eV to over 5.65 eV can be regulated by terminal ligand, providing tunable barrier height with semiconductor materials, which is a potentially important characteristic in Schottky-barrier based devices”

5. The band diagram in Figure 3h is confusing, V_g can adjust the channel layer work function and thus the barrier height between MXene and VBM is adjusted accordingly. Why do the authors give a fixed VBM of channel layer at -5.3 eV for both positive V_g and negative V_g operation?

“Author reply”: We are sorry for the misleading of band diagram. According to the metal-semiconductor contact theory, the energy level at the interface is fixed and the band bending occurs at the semiconductor layer when gate voltage is applied, which regulates Schottky barrier between metal and semiconductor. The -5.3 eV refers to the energy level at the contact interface and as shown in Figure 3h, and the VBM in the semiconductor is lower and higher than -5.3 eV after applying negative and positive gate voltage, respectively. To avoid the misleading, we drew a dash line to present the energy level of 5.3 eV at the interface, therefore, the regulation of Schottky barrier

after applying gate voltage can be more intuitively.

6. In Figure 2a, the authors only give device schematic, but no real images of the devices. The authors should provide real device images and give more details on the device size, such as channel layer thickness, contact pad size, the distance between contacts, and MXene layer conductivity.

“Author reply”: We thank reviewer for the valuable comments to improve our work. The real image of device is added in Fig. S4, and as shown in the microscope image, the distance between source and drain electrode is about 200 μm . To calculate the current density, the area of drain electrode is independent as about 200 \times 200 μm , and the thickness of PDVT-10 deposited on MXene and AgNWs is 76 nm and 125 nm, respectively. The difference of semiconductor thickness is ascribed to the hydrophilic nature of MXene and the plasma treatment of substrate before the spin coating of MXene. The conductivity of MXene and AgNWs layer are also measured, and due to the oxidation of MXene, the conductivity is relative smaller than AgNWs.

Supplementary Fig. 4. The microscope image of MVOFET.

Supplementary Fig. 12 The conductivity of (a) MXene and (b) AgNWs.

7. In the introduction part and later in the main text, the authors made several claims about the leakage current effect in the Vertical transistor, but the authors did not provide any experimental data on leakage currents. Since the leakage current is addressed as one of the key parameters, authors should give more analysis on it.

“Author reply” We thank reviewer for the valuable comments to improve our work. The leakage current mentioned in the main text referred to the current directly flew from the bottom source electrode to the top drain electrode, as shown in figure, which is induced by the vertical stacked of source and drain electrode. The output current of MVOFET reached saturation at about $V_{DS}=-6V$, while the output current of VOFET made by AgNWs increased with the increase of V_{DS} , indicating that there is no leakage current between source and drain electrode, which can be further proved by the COMSOL simulation shown in Figure 4. To further demonstrate the leakage current between gate and source electrode, the I_{GS} was also measured and added in Fig. S7, which exhibited low leakage current.

Supplementary Fig. 7. (a-b) The SS and leakage current of VOFET made by MXene and AgNWs, respectively.

8. From Fig. 1g, it is hard to say that MXene/PDVT-10 bilayer film has enhanced absorption in UV light based on such a slight difference.

“Author reply” We thank reviewer for the constructive comments, and we have deleted related sentence in revised manuscript

9. On page 8, Authors only tried MXene solution with 1, 2, 3 mg/ml and then claimed 3 mg/ml is the best. Any idea what happens at higher concentration? Is there a point where the concentration becomes too high to get a good performance?

“Author reply” We thank reviewer for the valuable comments to improve our work. Actually, we have tried higher concentration. However, as described in question 1, the gate control ability of VOFET is highly related with the shield effect, conductivity and the opening area of source electrode. A higher concentration resulted in a more continuous and thicker MXene film, which reduced the gate control ability as the shield effect of gate electric field was enhanced. On the other hand, the thickness and roughness of MXene film will increase with more stacked MXene flake, leading to higher off current. Therefore, the performance of device decreased when the concentration was too higher. The device performance under 4 mg/ml MXene was added in Fig. S6.

Supplementary Fig. 6. The transfer curves of MVOFET under different VDS with a concentration of (a) 1mg/ml, (b) 2 mg/ml and 4 mg/ml MXene, respectively.

10. On page 9, the authors claim that “SEM and AFM images of AgNWs deposited on SiO₂ substrate with a concentration of 0.5 mg/ml (optimized concentration)” Any data to support this optimization?

“Author reply” We thank reviewer for the valuable comments to improve our work. The device performance with different concentration of AgNWs is shown below. Due to the large diameter (about 30 nm) of AgNWs, the thickness of and the shield effect of stacked AgNWs increase with the increase of concentration, which reduce the gate control ability of device and easily lead to the conduction of the source and drain electrodes. While the continuity and conductivity decrease when the concentration of AgNWs is lower, leading to low on state current and even open circuit of source and drain electrode. 0.5 mg/ml is the optimized concentration according to our experiment results, which is also consistent with our previous works. Nano Energy 2021, 85, 106010, ACS applied materials & interfaces 2018, 10 (36), 30587-30595 and ACS Photonics 2018, 5 (9), 3712-3722.

11. In Figure 3g, the authors only give the UPS data of MXene. Why not also give the AgNWs UPS data?

“Author reply” We thank reviewer for the valuable comments to improve our work. Silver nanowires are common mesh sources in vertical transistors, and their work function has also been reported. The work function in this work is consistent with reported literature Composites Part A: Applied Science and Manufacturing 139, 106088 (2020), ACS applied materials & interfaces 7, 2149-2152 (2015), Nano Energy 16, 122-129 (2015), and Adv Funct Mater 27, 1703541 (2017).

12. On page 12, the authors claim that “The superior conductivity and small sheet resistance endowed MXene unparalleled carrier transport speed.” The authors should show the experimental evidence the MXene has superior conductivity than Ag NW.

“Author reply” We thank reviewer for the careful and constructive comments to improve our work. The conductivity of MXene and 0.5 mg/ml based AgNWs was provided in the revised manuscript. Due to the oxidation, the conductivity of MXene film is smaller than AgNWs, which is the main reason for the relative smaller on state current of MVOFET. To ensure the rigor of the article, we have modified our statement in the revised manuscript. However, as described in question 1, the main factor that makes MXene better than AgNWs is not because of its conductivity. It is the excellent gate control ability of MVOFET (possess two models for regulating Schottky barriers in a single material system), the compatibility with semiconductor due to tunable working function and other excellent intrinsic properties of MXene such as ultra-thin thickness, hydrophilicity and large-scale ability.

Supplementary Fig. 12 The conductivity of (a) MXene and (b) AgNWs.

13. Simulation in Figure 4 is the kind of planer transistor with symmetrical MXene or Ag NW contact in both source and drain. However, the paper is focused on vertical transistors with asymmetrical MXene contact only in the source area. Please comment on this.

“Author reply” We thank reviewer for the valuable comments to improve our work. Actually, the simulation in Figure 4 is vertical transistor and not planer transistor. The simulation structure is the cross section of vertical transistor at the perforation of source electrode. The interface between source electrode and semiconductor layer of vertical transistors are formed by the staked MXene flakes or AgNWs and the openings between them. Take a perforation as example, it is surrounded by MXene flakes or AgNWs, and they are all served as source electrode. Therefore, in the view of cross section at the perforation, it is like a bottom contact planer transistor. However, the drain electrode is above the semiconductor, the source electrode and drain electrode are vertically stacked in space. There are lots of perforations in the vertical transistor, and we adopted the basic structure in Figure 4a as a simplification.

14. In Figure 5, the semiconducting polymer and MXene are covered by opaque gold electrodes, so how can they detect light?

“Author reply” We thank reviewer for the valuable comments to improve our work. As shown in the microscope image of MVOFET, the semiconductor layer and MXene are not all covered by top drain electrode, and some part of semiconductor and MXene are exposed. Besides, the light strikes the device at an oblique angle, therefore, the semiconductor layer can also generate photogenerated carriers. Certainly, the performance of photodetector can be further enhanced utilizing transparent drain electrode. Similar vertical photodetector transistor structure with opaque electrodes has also been reported in Adv. Mater. 30, e1803655 (2018), Journal of Materials Chemistry C 8, 12632-12637 (2020), Opt. Mater. 100, 109664 (2020) and ACS Photonics 5, 3712-3722 (2018).

Supplementary Fig. 4. The microscope image of MVOFET.

15. What about the hysteresis in these devices? Have the authors studied how the stability of the devices with respect to hysteresis and threshold voltage shift with the

bias? This is important information should be carefully analyzed.

“Author reply” We thank reviewer for the careful and constructive comments to improve our work. The hysteresis under double sweep and the bias stability of MVOFET and AgNWs based VOFET are provided in Fig. S18. As shown in Fig. S18, the ΔV_{TH} of MVOFET is two times smaller than AgNWs based VOFET, indicating that the interface trap density of MVOFET is smaller than AgNWs. This can be ascribed to the small roughness of MXene and the clean interface between semiconductor layer.

Supplementary Fig. 18. The double sweep measurement of (a) MXene and (b) AgNWs.

The bias stability of MVOFET was provided in Fig. S9. As shown in Fig. S9, a small right shift of threshold voltage occurred when a positive gate voltage was applied and the ΔV_{TH} increased with the increase of bias time. The shift of threshold voltage is on account of the accumulation and depletion of carriers in the perforation of MXene film. The device bias stability can be improved by reducing the perforation of MXene, however this will reduce the gate control ability and hinder the device performance. As shown in Fig. 3i, the bias stability is significantly enhanced along with the decrease of device performance when the MXene film is without perforation

Supplementary Fig. 9. The bias stability of MVOFET measured at 5V gate voltage

with different bias time

16. The title use of the term “MXetronics” here is not really appropriate. Please remove it. A more appropriate title maybe “MXene Based Saturation Organic Vertical Photoelectric Transistors with Low Subthreshold Swing”

“Author reply” We thank reviewer for the valuable comments to improve our work. The title has changed into “MXene Based Saturation Organic Vertical Photoelectric Transistors with Low Subthreshold Swing”

17. There are few claims that the authors make which don't have any supporting data or seem like an opinion:

a. The authors mentioned graphene is complicated because of CVD growth. Actually, there are many reported solution-processed graphene.

“Author reply” We thank reviewer for the valuable comments to improve our work. The graphene indeed can be prepared by solution process, however, to serve as the source electrode of VOFET, the monolayer graphene is usually required. Therefore, to ensure the performance of VOFET, vertical transistors have been fabricated usually from pristine graphene electrodes prepared through mechanical exfoliation or thermal chemical vapor deposition and further transferred to substrate as the source electrode of vertical transistor (Adv. Mater. 2016, 28, 4803–4810, Adv. Funct. Mater. 2019, 29, 1808453, Science China-Information Sciences 2020, 63, 201401). The rGO can be used as solution-process source electrode of VOFET, however, complicated reduction process of GO such as chemical reduction or high temperature reduction are needed instead, resulting in poor transistor performance (the on/off ration can only reached 10^3 , Chemistry of Materials 2018, 30 (3), 636-643.). To ensure the rigor of the paper, the statement is deleted in revised manuscript.

b. The authors claimed that AgNWs are unfavorable for achieving high-performance vertical transistors due to the large sheet resistance in the introduction part. AgNWs are very conductive and maybe even better than unoxidized MXene { Nano Futures 4 035002, (2020) Composites Part A: Applied Science and Manufacturing 149, 106545 (2021) }..

“Author reply” We thank reviewer for the valuable comments to improve our work. The conductivity of MXene film and 0.5 mg/ml AgNWs are measured and compared in Fig. s12. The detail description was provided in question 12. The main factor that makes MXene better than AgNWs is not because of its conductivity. It is the excellent gate control ability of MVOFET (possess two models for regulating Schottky barriers in a single material system), the compatibility with semiconductor due to tunable working function and other excellent intrinsic properties of MXene such as ultra-thin thickness, hydrophilicity and large-scale ability.

Supplementary Fig. 12 The conductivity of (a) MXene and (b) AgNWs.

c. the authors claim that “there are very few reports of the application of MXene in organic transistors” There are already several papers {Adv. Mater. 2021, 33, 2008215; ACS Appl. Mater. Interfaces 2020, 12, 29, 32970–32978; ACS Nano 2019, 13, 10, 11392–11400; Microchimica Acta volume 188, Article number: 301 (2021) Nanoscale, 2018,10, 5191-5197}. Again, the investigation in the background is not enough.

“Author reply” We thank reviewer for the valuable comments to improve our work. Indeed, there are reported works about the application of MXene in organic transistors. However, they are served as a blend material of semiconductor or the electrode of planner transistor, which is totally different with our work. The novelty in this work is the first to utilize MXene as the source electrode of vertical transistor and improve the performance of vertical transistors. Moreover, its underling mechanism is totally different. To ensure the rigor of the article, we have modified our statement in the revised manuscript.

Reviewer #2 (Remarks to the Author):

This paper introduces the transition-metal carbides/nitrides (MXene) material as a new source electrode of organic vertical transistors, which improves the subthreshold swing (SS) to 73 mV/dec comparing with the 350 mV/dec of traditional AgNWs based vertical transistors. This application also takes advantage from the ultra-thin thickness and oxidation properties of MXene and greatly improves the saturation of output current. The experiment is explained in detail and finite element modeling cases are conducted in COMSOL Multiphysics. The modeling results support and validate their theory. This innovative application is promising and expands the application of MXene materials in the fields of electronic devices. However, there are some major problems or flaws to be addressed before being further consideration for publication.

1, In the introduction section, the electro-chemical and mechanical property backgrounds of MXene material and why it is specifically suitable for serving such purposes are not well established. This lack of explanation fails to illustrate the necessity of using this class of material rather than other 2D materials.

“Author reply” Thank you for the valuable advice of reviewer to improve our work. According to the complementary experiments, we are surprised to find that the MXene take advantage of both graphene and mesh metal source electrode. The ultra-thin film of MXene with perforation probes both schottky barrier modulation models of graphene and mesh metal electrode, which significantly improve the gate control ability of MVOFET resulting in the small SS. To further highlight the superiority of MXene as a mesh electrode of VOFET, the introduction was re-organized and the superiority of MXene was discussed below.

Firstly, the operation of VFET relies on the gate-tunable modulation of Schottky barrier in the interface between source electrode and semiconductor and thus regulated the injection of carriers. Depending on the utilized of source electrode, the working mechanisms of VFET can be roughly divided into two categories. A class is the continuous single layer graphene and another is the mesh metal electrode. In the graphene based VFET, the graphene just partially screened the gate field, therefore, the gate field can penetrate graphene and then modulate the Schottky barrier. To reduce the shielding effect, single layer graphene is usually needed. However, the gate field is still screened to some extent hindering the gate control ability of graphene based VFET. Except for graphene, there are some VFETs using mesh metal electrode as source electrode. The gate field will be screened by conventional metal such as AgNWs. Therefore, to function those devices, the metallic source electrode is required to be perforated, thus allowing the gate field direct access to the metal-semiconductor interface, where a field induced band bending lowers the barrier to allow channel currents. The porous structure of source electrode reduces the shielding effect of the gate electric field and improves the gate control capability of the device. The MXene film possesses a ultra-thin thickness of about 1 nm which is similar to the single layer graphene. Therefore, we envisage that the gate electric field can also pass through the ultra-thin MXene film to further regulator the Schottky

barrier between MXene and semiconductor. To verify this assumption, we constructed a continuous monolayer MXene film without perforation utilizing a lateral self-assembly method as described in ACS Nano 2021, 15, 625–636.

Fig. 3. The working mechanism of MVOFET and AgNWs based VOFET. (a) The output characteristic of MVOFET under 313K and 353 K. (b) $\ln(I_{SAT}/T^2)$ versus $q/K_B T$ plots at various gate voltages. (c) The variation of Schottky height under different gate voltage. (d) The output characteristic of AgNWs based VOFET under 313K and 353 K. (e) $\ln(I_{SAT}/T^2)$ versus $q/K_B T$ plots at various gate voltages of AgNWs based VOFET. (f) The variation of Schottky height under different gate voltage of AgNWs based VOFET. (g) The microscope image of stacked MXene flakes without perforation. (h) 10 cycles of transfer curves of MVOFET based on the stacked single layer MXene flakes without perforation. (i) The bias stress ability of the device (j) UPS measurement of MXene film. (k) Energy-band diagram of

MVOFET and AgNWs based VOFET.

Fig.3g, is the microscope image of stacked MXene flakes without perforation. Fig. 3h shows the 10 cycles of transfer curves of MVOFET based on the stacked single layer MXene flakes without perforation and Fig. 3i is the bias stress ability of the device. Compared with the device construct by stacked MXene with perforation, the device without perforation exhibits larger off state current, relative poor gate control ability (large SS) while an improvement of device stability. The device exhibits typical p-type transistor curves with an on/off ratio over 10^3 , indicating that the channel current can be regulated by gate voltage even without perforation. These experimental phenomenon shows that the continuous MXene film just partially screens the gate electric field, while the residual gate electric field penetrates MXene and modulates Schottky barrier, which is similar to single layer graphene. The relative poor SS is ascribed to the shield of gate electric field. The device stability of MVOFET with perforation is relative poor than the device without perforation, which can be attributed to the accumulation and depletion of carriers in the perforation resulting in the shift of threshold voltage. Therefore, the MVOFET with perforation can probe both modes (graphene and mesh metal electrode) in a single material system, which significantly improves the gate control capability of the device.

Secondly, compared with AgNWs, the work function of MXene is higher, which reduces the injection barrier of carriers. Besides, the variation of equivalent SB height of AgNWs based VOFET measured in Figure 3 is smaller than that of MVOFET under the same gate voltage, indicating a stronger gate tunability in MVOFET. Moreover, unlike the rod-shaped AgNWs, the 2D flake structure of MXene possesses much smaller roughness (1.81 nm of MXene and 11.7 of AgNWs under the same range). The variation of threshold voltage (ΔV_{TH}) measured from the hysteresis for double sweep of AgNWs and MVOFET is provided in Fig. S18. As shown in Fig. S18, the ΔV_{TH} of MVOFET is two times smaller than AgNWs based VOFET. Therefore, the MXene provides excellent interface with semiconductor layer, which further reduces the SS of MVOFET. The basic transistor performance of MVOFET with graphene, AgNWs based VOFET and other reported vertical organic transistors are compared in Table S1 to show the superior property of MXene itself.

Semiconductor Materials	Source electrode	SS	Ion/Ioff	Current density	Threshold voltage(V)	Ref
C ₆₀	Grahphene	>2(V/dec) ^a	3×10^3	1×10^{-6} (A)	>50 ^a	1
PBDB-T	Grahphene	>2(V/dec) ^a	10^4	1×10^{-6} (A)	4	2
PC ₇₁ BM	Grahphene	>2(V/dec) ^a	10^5	1×10^{-5} (A)	-5	2
C ₆₀	Grahphene	5.75(V/dec)	8×10^4	3×10^{-6} (A)	-10	3
PTCDI-C ₈	Grahphene	>2(V/dec) ^a	< 10^3	12.4 (mA/cm ²)	>-20 ^a	4

Pentacene	Grahphene	$>2(\text{V}/\text{dec})^a$	$<10^3$	10.5 (mA/cm ²)	$>30^a$	4
DNTT	CNT	500(mV/dec)	10^5	110 (mA/cm ²)	1	5
MoS ₂	CNT	$>2(\text{V}/\text{dec})^a$	10^3	1×10^{-6} (A)	$>40^a$	6
MAPbI ₃	Porous ITO	1.1(V/dec)	10^4	1 (mA/cm ²)	1	7
CuPc	Pattern Au	$>2(\text{V}/\text{dec})^a$	10	0.5 (A/cm ²)	1	8
PDVT-8	AgNWs	500(mV/dec)	2×10^4	6.5 (mA/cm ²)	1.5	9
PDVT-10	Graphene	1.75(V/dec)	1×10^4	37.5(mA/cm ²)	8.5	This work
PDVT-10	AgNWs	350(mV/dec)	1×10^5	9.2 (mA/cm ²)	0.7	This work
PDVT-10	MXene	73 (mV/dec)	2×10^5	5.8 (mA/cm ²)	1.2	This work

^a estimated from the transfer curves of reported devices.

Supplementary Fig. 18. The double sweep measurement of (a) MXene and (b) AgNWs.

Thirdly, the solution processable property of MXene endows MVOFET the feasibility of large-scale fabrication and the adjustable working function (terminal ligand engineering) allows MVOFET more compatible with organic materials. The issue of the unsaturation of output current in VOFET is solved with MXene due to its ultra-thin thickness and native oxidation. Besides, the native oxidation of MXene broadens the detection spectrum of VOFET without additional layer and the ultra-thin thickness would offer great prospects in ultra-thin and 2D semiconductor based VOFET. Therefore, the utilization of MXene as the source electrode of VFET not only improves the device performance and solves the unsaturation issue of VFET but also offers great potential in organic electronics. Relative discussion has been added in the main manuscript.

2, Throughout this paper, no molecular dynamics or density functional theory models are being conducted to explore the electrical and mechanical properties of the MXene in the electrode application, which would provide insights to the nanoscale. The finite element modeling using COMSOL Multiphysics alone is not enough to serve as theoretical background due to the scale of the experiment.

“Author reply” Thank you for the valuable advice of reviewer to improve our work. Limited by the experiment conditions, the molecular dynamics or density functional theory models are not conducted in this work. However, the electrical and mechanical properties of the MXene have been widely studied in previous works through the density functional theory, and the simulation results are consistent with our experimental results. So we do not conduct DFT in this work. As described in the reference of J. Mater. Chem. A, 2021, 9, 5016–5025, the pristine MXene, $\text{Ti}_3\text{C}_2\text{T}_x$, exhibits strong metallic conductivity. After light oxidation of $\text{Ti}_3\text{C}_2\text{T}_x$, the MXene still keeps the metallic properties and the working function of MXene increases, which is consistent with the UPS and XPS measurement of our work. The MXene still keeps the metallic conductivity while the XPS reveals the existence of TiO_2 . Besides, this also supports the experimental results presented in the SEM images, which showed TiO_2 nanoparticles on the surface of flakes, while still maintained conductive MXene layer. After heavy oxidation, the $\text{Ti}_3\text{C}_2\text{T}_x$ is almost completely transformed into TiO_2 and loses the metallic conductivity. The reference of EPL (Europhysics Letters) 2015, 111 (6), 67002 and Chemistry of Materials 2019, 31 (17), 6590-6597 also demonstrated the electrical and mechanical properties of $\text{Ti}_3\text{C}_2\text{T}_x$ revealing the working function and metallic conductivity of MXene.

3, Too few samples are discussed in this work, and this reduces the generosity of the application. More experimental results are recommended to provide a statistic view of the essential properties of MXene VOFET.

“Author reply” Thank you for the valuable advice of reviewer to improve our work. We have tested 30 MVOFET and the SS, current density and Ion/Ioff ratio were statisticed in the distribution histogram (Fig. S8). Due to the inhomogeneity of the MXene solution and the randomness of the MXene distribution during spin coating, different devices have certain performance differences. According to the distribution histogram of 30 MVOFET devices, the SS was mostly distributed between 70-90 mv/dec, which was still much smaller than AgNWs based VOFET. The distribution of current density and Ion/Ioff ratio were also counted which exhibited acceptable device variation.

Supplementary Fig. 8. The distribution histogram of basic transistor performance under 30 MVOFET devices. The distribution of (a) SS, (b) current density and (c) I_{on}/I_{off} of MVOFET.

4, The grammar and the spelling are recommended to be checked thoroughly to eliminate existing typos. Short straight sentences are recommended to replace the long complicate ones to help with the understanding.

“Author reply” Thank you for the valuable comments of reviewer, and the grammar and the spelling are thoroughly checked in the revised manuscript. Short sentences are used to replace the long complicate one.

Based on the reviews, I do not recommend the paper to be accepted with the current form.

Reviewer #3 (Remarks to the Author):

The authors have used MXene as the electrode for the vertical photoelectric transistors, showing a low subthreshold swing value of 73 mV/dec. The whole content may be suitable for publication in this journal if the authors can provide proper answers and corresponding revision to comments listed below,

1. It is unfair to compare the MXene 2D materials with silver nanowire with a high surface roughness that will degrade the device performance. It is predictable that the subthreshold swing of MXene based devices will be lower than that of AgNW based devices. The reviewer strongly recommends the authors compare MXene with graphene with similar conductivity to show the superior property of MXene itself.

“Author reply” Thank you for the valuable comments of reviewer. The device performance of graphene based VOFET was added in Fig. S13. The device performance is comparable with reported works, however, due to the shield effect of graphene, the SS and Ion/Ioff of device is still large than our MVOFET device. To get a comprehensive comparison, we compared the basic transistor performance of MVOFET with other reported VOFET including graphene and other source electrodes to show the superior property of MXene itself. The basic transistor performance comparison between MVOFET and other vertical organic transistor are summarized in Table S1. As shown in Table S1, the device performance of this work is superior to other reported vertical transistor, especially in the SS of device.

Semiconductor Materials	Source electrode	SS	Ion/Ioff	Current density	Threshold voltage(V)	Ref
C ₆₀	Grahphene	>2(V/dec) ^a	3 × 10 ³	1 × 10 ⁻⁶ (A)	>50 ^a	1
PBDB-T	Grahphene	>2(V/dec) ^a	10 ⁴	1 × 10 ⁻⁶ (A)	4	2
PC ₇₁ BM	Grahphene	>2(V/dec) ^a	10 ⁵	1 × 10 ⁻⁵ (A)	-5	2
C ₆₀	Grahphene	5.75(V/dec)	8 × 10 ⁴	3 × 10 ⁻⁶ (A)	-10	3
PTCDI-C ₈	Grahphene	>2(V/dec) ^a	<10 ³	12.4 (mA/cm ²)	>-20 ^a	4
Pentacene	Grahphene	>2(V/dec) ^a	<10 ³	10.5 (mA/cm ²)	>30 ^a	4
DNTT	CNT	500(mV/dec)	10 ⁵	110 (mA/cm ²)	1	5
MoS ₂	CNT	>2(V/dec) ^a	10 ³	1 × 10 ⁻⁶ (A)	>-40 ^a	6
MAPbI ₃	Porous ITO	1.1(V/dec)	10 ⁴	1 (mA/cm ²)	1	7
CuPc	Pattern Au	>2(V/dec) ^a	10	0.5 (A/cm ²)	1	8

PDVT-8	AgNWs	500(mV/dec)	2×10^4	6.5 (mA/cm ²)	1.5	9
PDVT-10	Graphene	1.75(V/dec)	1×10^4	37.5(mA/cm ²)	8.5	This work
PDVT-10	AgNWs	350(mV/dec)	1×10^5	9.2 (mA/cm ²)	0.7	This work
PDVT-10	MXene	73 (mV/dec)	2×10^5	5.8 (mA/cm ²)	1.2	This work

^a estimated from the transfer curves of reported devices.

Supplementary Fig. 13 The device performance of graphene based VOFET. (a) and (b) The transfer curves and output current of graphene based VOFET.

2. In Supplementary Fig. 1, due to the large particles, the surface roughness seems much larger than 1.47 nm as described in line 151.

“Author reply” Thank you for the valuable comments of reviewer. The surface roughness indeed increased with the range of AFM image due to the oxidation of MXene. However, we compared the roughness of MXene film and AgNWs in the same range, the results and the conclusion are still important. Limited by the test instrument, we cannot provide a large range AFM image of MXene film, therefore we provide a AFM image with more stacked MXene flakes in the revised manuscript. As shown in the AFM image, the roughness of MXene film is 1.81 and the thickness of single layer MXene flake is 1.7 nm, compared with the AFM image of AgNWs measured in the same range, the roughness of MXene film is still much smaller than AgNWs (11.7). Besides, we have provided a microscope image of MXene film with large range, which also demonstrated the uniform deposition of MXene film, therefore, the overall roughness of MXene film will not increase too much. So we can still conclude that MXene has much better interface quality compared with Ag NWs. Besides, based on our new experiments, the interface quality is not the major reason for the small SS of MVOFET. The MXene take advantage of both graphene and mesh metal source electrode, resulting in the strong gate control ability of device.

Fig. 1. Structure of semiconductor and characteristics of MXene film. (a) The chemical structure of PDVT-10, DPP-DTT and Pentacene. (b) SEM image of MXene film with a concentration of 3 mg/ml. (c) TEM image of dispersive MXene nano flake. (d) HRTEM image of MXene. (e) Microscope image of MXene film. (f) The AFM image of stacked MXene. (g) The UV-vis absorption spectrum of MXene and PDVT-10.

3. is it possible to obtain an MXene film of a d-spacing of 1.25 nm without any post-treatment? Please refer to the following paper. Science 2020, 369, 446.

“Author reply” Thank you for the valuable comments of reviewer. We have checked the manuscript and found no statement with MXene film of a d-spacing of 1.25 nm. To confirm the d-spacing of MXene, we have checked the measurement of DigitalMicrograph utilizing FF model and confirmed that the d-spacing of MXene film is 0.27 nm (1.35 nm), which is consistent with the reference of Science 2020, 369, 446. The reference was cited in revised manuscript.

4. The distribution of TiO₂ is non-uniformly MXene as a particle instead of a film. Hence the reproducibility might be below. Please show a histogram of the device's performance to show reproducibility.

“Author reply” Thank you for the valuable comments of reviewer. We have tested 30 MVOFET and the SS, current density and Ion/Ioff ratio were statisticed in the distribution histogram (Fig. S8). Due to the inhomogeneity of the MXene solution and the randomness of the MXene distribution during spin coating, different devices have certain performance differences. According to the distribution histogram of 30

MVOFET devices, the SS was mostly distributed between 70-90 mv/dec, which was still much smaller than AgNWs based VOFET. The distribution of current density and I_{on}/I_{off} ratio were also counted which exhibited acceptable device variation.

Supplementary Fig. 8. The distribution histogram of basic transistor performance under 30 MVOFET devices. The distribution of (a) SS, (b) current density and (c) I_{on}/I_{off} of MVOFET.

5. Please show the change in device performance with the variation of TiO₂ particle density.

“Author reply” Thank you for the valuable advice of reviewer to improve our work. The device performance of different concentration of MXene and the device performance with different ALD TiO₂ thickness can reflect the device performance with different TiO₂ particle density. With the increase of TiO₂ particle density, more electrons can be trapped in the TiO₂ particle due to the difference of energy level of PDVT-10 and TiO₂, resulting in a right shift of threshold voltage. Besides, the conductivity of MXene decreases with the degree of oxidation, as a consequence, the on state current decreases with the increase of TiO₂ particle.

Supplementary Fig. 20. The device performance of MVOFET with different TiO₂ thickness. (a) The transfer curves of MVOFET with different thickness of TiO₂ (b) The output current of MVOFET with 1 nm TiO₂.

6. Please use solution-processed or sputter TiO₂ layer with different thicknesses as a comparison,

“Author reply” Thank you for the valuable advice of reviewer to improve our work. In order to avoid the influence of organic solvents and high-energy particle sputtering,

we used atomic layer deposition (ALD) to fabricate different thickness of TiO_2 film instead. As shown in Fig. S19, after the deposition of TiO_2 film with different thickness, the devices exhibit typical p-type transistor performance. However, the current density decreased along with a right shifted of threshold when the thickness of TiO_2 increased. Fig. S19 was the output current of MVOFET with a 1 nm TiO_2 layer, indicating that the saturation of output current of VOFET can be achieved by depositing an oxide insulating layer above the MXene source electrode. Therefore, the saturation mechanism of the output current of MVOFET is proved from the perspective of theoretical simulation and experiment.

7. Device simulation is lateral structure, but the actual device is vertical. Please use a similar device structure.

“Author reply” Thank you for the valuable advice of reviewer to improve our work. Actually, the simulation in Figure 4 is vertical transistor no planer transistor. The simulation structure is the cross section of vertical transistor at the perforation of source electrode. The interface between source electrode and semiconductor layer of vertical transistors are formed by the staked MXene flakes or AgNWs and the openings between them. Take a perforation as example, it is surrounded by MXene flakes or AgNWs, they are all served as source electrode. Therefore, in the view of cross section at the perforation, it is like a bottom contact planer transistor. However, the drain electrode is above the semiconductor, the source electrode and drain electrode are vertically stacked in space. There are lots of perforations in the vertical transistor, and we adopted the basic structure in Fig. 4a as a simplification.

8. In addition, TiO_2 did not form a layer of 2 nm in the actual device.

“Author reply” Thank you for the valuable advice of reviewer to improve our work. Actually, the thickness of TiO_2 cannot be calculated and the TiO_2 did not form a layer of 2 nm in real device. In order to more intuitively reflect the role of the oxidation of MXene and the convenience of simulation experiments, we utilized a 2 nm TiO_2 instead. According to the simulation and the experiment results in question 6, an oxidation layer can reduce the leakage current between source and drain electrode and realize the saturation of output current. While the oxidation layer can also reduce the on state current of device, which is also consistent with the experiment phenomenon of MVOFET in this work.

9. In line 305, please explain the faster response of MVOFET to UV light than to visible light.

“Author reply” Thank you for the valuable advice of reviewer to improve our work. The faster response of UV light to visible light can be ascribed to the strong absorption of oxidation MXene in the UV wavelength as shown in Figure 1i. Besides, the energy of UV light is stronger than visible light which excites more photogenerated carriers in TiO_2 particle. Thirdly, the large HOMO level difference between TiO_2 and PDVT-10 leads to the quick transfer of photo-generated holes to PDVT-10 layer.

10. Please compare the device performance using MXene with other vertical photoelectric transistors from the previous results and summarize it in one figure to show the significance of this work.

“Author reply” Thank you for the valuable advice of reviewer to improve our work. We have already compared the photodetector performance of MXene with other planar and vertical photoelectric transistors in Table S2. To make a better comparison with other vertical photoelectric transistors, we have added more reported vertical photoelectric transistors in Table S2 in the revised manuscript. As shown in the table, the formation of TiO₂ particle broadens the range of detection without additional function layer. Besides, the basic photodetection performance is comparable and even better than planar and vertical based photodetector devices.

Table S2. The comparison between MVOFET and other reported phototransistors.

Structure	Materials	R (A/W)	D (Jones)	I _{light} / I _{dark}	λ(nm)	Speed (s)	Ref
Planner	DPP-DTT/CsPbI ₃	110	2.9×10 ¹³	6×10 ³	350–940	3.2	11
Planner	C60	8×10 ⁻³	N/A	10	350-650	1.8	12
Planner	Graphene oxide	1×10 ⁻²	N/A	N/A	400-1600	0.018	13
Planner	SnS ₂	860	1.1×10 ¹⁰	<10	300-750	0.7	14
Planner	2F-4-TFPTA	3.6×10 ³	N/A	10	465-622	0.043	15
	Single crystal						
Planner	AZO/ ZnO NP	8×10 ⁻²	N/A	80	365	0.11	16
Vertical	CsPbBr ₃ NCs	2.2×10 ³	1.1×10 ⁹	N/A	405	0.02	17
	/Graphene						
Vertical	PbSe QDs	28	1.3×10 ¹³	10 ⁴	1064	0.11	18
	/AgNWs						
Vertical	DPA single crystal	110	1×10 ¹³	10 ⁴	420	>1	19
	/Graphene						
Vertical	PDVT-8/AgNWs	150	1.3×10 ¹¹	2×10 ³	400-700	2	9
Vertical	PTCDI-C ₈ and	0.4	1.2×10 ⁹	10 ⁴	480-670	0.1	20
	Pentence/Graphene						
Vertical	PDVT-10/MXene	366	2.8×10 ¹²	7×10 ⁴	365-700	0.01	This work

REVIEWERS' COMMENTS

Reviewer #1 (Remarks to the Author):

The authors have addressed my comments in detail. The manuscript may now be accepted. I suggest the English should be further improved.

Reviewer #3 (Remarks to the Author):

In this manuscript, authors have proposed the MXene that take advantage of both graphene and metal mesh electrode as the source electrodes for the vertical transistors. The authors have precisely revised the manuscript as the reviewer suggested. Therefore, the reviewer strongly recommends publishing this manuscript as it is without any further revision.

We would like to thank all the reviewers for their critical suggestions and valuable comments. Reviewer's comments are in blue, while our responses are immediately below. Modifications to the manuscript are highlighted in the manuscript itself.

Reviewer #1.

The authors have addressed my comments in detail. The manuscript may now be accepted. I suggest the English should be further improved.

Author reply: We thank reviewer for the careful and constructive comments to improve our work. We have carefully checked our manuscript and the English is further improved. The modifications to the manuscript are highlighted in the revised manuscript.

Reviewer #3

In this manuscript, authors have proposed the MXene that take advantage of both graphene and metal mesh electrode as the source electrodes for the vertical transistors. The authors have precisely revised the manuscript as the reviewer suggested. Therefore, the reviewer strongly recommends publishing this manuscript as it is without any further revision.

Author reply: We are very grateful to the reviewers for their recognition of our work.

Peer Review File

Title: MXene based Saturation Organic Vertical Photoelectric Transistors with Low Subthreshold Swing